# Spin-qubit control with a milli-kelvin CMOS chip

Samuel K. Bartee[1,6], Will Gilbert[2,3], Kun Zuo[1], Kushal Das[1,4], Tuomo Tanttu[2,3], Chih Hwan Yang[2,3], Nard Dumoulin Stuyck[2,3], Sebastian J. Pauka[1,4], Rocky Y. Su[3], Wee Han Lim[2,3], Santiago Serrano[2,3], Christopher C. Escott[2,3], Fay E. Hudson[2,3], Kohei M. Itoh[5], Arne Laucht[2,3], Andrew S. Dzurak[2,3] & David J. Reilly[1,4 ✉]

A key virtue of spin qubits is their sub-micron footprint, enabling a single silicon chip to host the millions of qubits required to execute useful quantum algorithms with error correction[1–3]. However, with each physical qubit needing multiple control lines, a fundamental barrier to scale is the extreme density of connections that bridge quantum devices to their external control and readout hardware[4–6]. A promising solution is to co-locate the control system proximal to the qubit platform at milli-kelvin temperatures, wired up by miniaturized interconnects[7–10]. Even so, heat and crosstalk from closely integrated control have the potential to degrade qubit performance, particularly for two-qubit entangling gates based on exchange coupling that are sensitive to electrical noise[11,12]. Here we benchmark silicon metal-oxide-semiconductor (MOS)-style electron spin qubits controlled by heterogeneously integrated cryo-complementary metal-oxide-semiconductor (cryo-CMOS) circuits with a power density sufficiently low to enable scale-up. Demonstrating that cryo-CMOS can efficiently perform universal logic operations for spin qubits, we go on to show that milli-kelvin control has little impact on the performance of single- and two-qubit gates. Given the complexity of our sub-kelvin CMOS platform, with about 100,000 transistors, these results open the prospect of scalable control based on the tight packaging of spin qubits with a 'chiplet-style' control architecture.

Utility-scale quantum computing probably requires millions of physical qubits, operated by auxiliary classical systems that generate more than a trillion control signals per second[13,14]. In realizing this vast and complex platform, silicon qubits present advantages with their small footprint[15], long coherence times[16] and inherent compatibility with VLSI (very large scale integrated) control circuits. Although the potential for integrated control has been a key motivator for the progress in silicon-based qubits over the past two decades, so far this aspect has remained largely undeveloped.

Despite the advantages of integrated control[4,5], a serious concern arises from the heat and crosstalk generated by modern complementary metal-oxide-semiconductor (CMOS) circuits. In relation to heat, this problem is eased by recent work showing that spin qubits continue to function at elevated temperatures[17–19]. But two-qubit entangling gates remain sensitive to electrical noise[11,12], arising, for instance, from volt-scale, sub-nanosecond switching of proximal CMOS transistors. One way of partially mitigating these adverse effects is to separate the control system to 4 K, connecting to milli-kelvin qubits using long cables[8,20]. Cable connectivity poses an additional barrier to scaling up the control interface[4], given the extreme density of interconnects required to operate even modest numbers of qubits.

Here we demonstrate the control of MOS-style silicon spin qubits using a heterogeneously integrated cryo-CMOS chip operating at milli-kelvin temperatures, as shown in Fig. 1a. Heterogeneous, 'chiplet-style' integration, as opposed to monolithic circuits, decouples the hot and noisy control system from sensitive qubits and retains the potential for dense, lithographically defined chip-to-chip interconnects needed to manage the wiring challenge inherent to spin qubits. We demonstrate that this chiplet architecture supports a control scheme that leverages a global resonance field to enable complete universal control of spin qubits using the baseband pulses that can be generated efficiently with proximal, low-power cryo-CMOS.

The details of the CMOS control chip have been reported previously with an early conceptual demonstration using GaAs quantum dot structures[7]. The effect of milli-kelvin CMOS on qubit performance, however, has remained an open question until the present work. As the spin degree of freedom is decoupled from electrical noise, integrated CMOS is expected to have only a minor impact on single-qubit operations. By contrast, coupling spins by Heisenberg exchange creates the most sensitive probe of voltage noise known[11,12] because the exchange energy can depend exponentially on gate voltage. Countering this intuition, we show that even for noise-sensitive two-qubit gates, our

[1]ARC Centre of Excellence for Engineered Quantum Systems, School of Physics, The University of Sydney, Sydney, New South Wales, Australia. [2]Diraq, Sydney, New South Wales, Australia. [3]School of Electrical Engineering and Telecommunications, University of New South Wales, Sydney, New South Wales, Australia. [4]Emergence Quantum, Sydney, New South Wales, Australia. [5]School of Fundamental Science and Technology, Keio University, Yokohama, Japan. [6]Present address: Diraq, Sydney, New South Wales, Australia. ✉e-mail: David.Reilly@sydney.edu.au

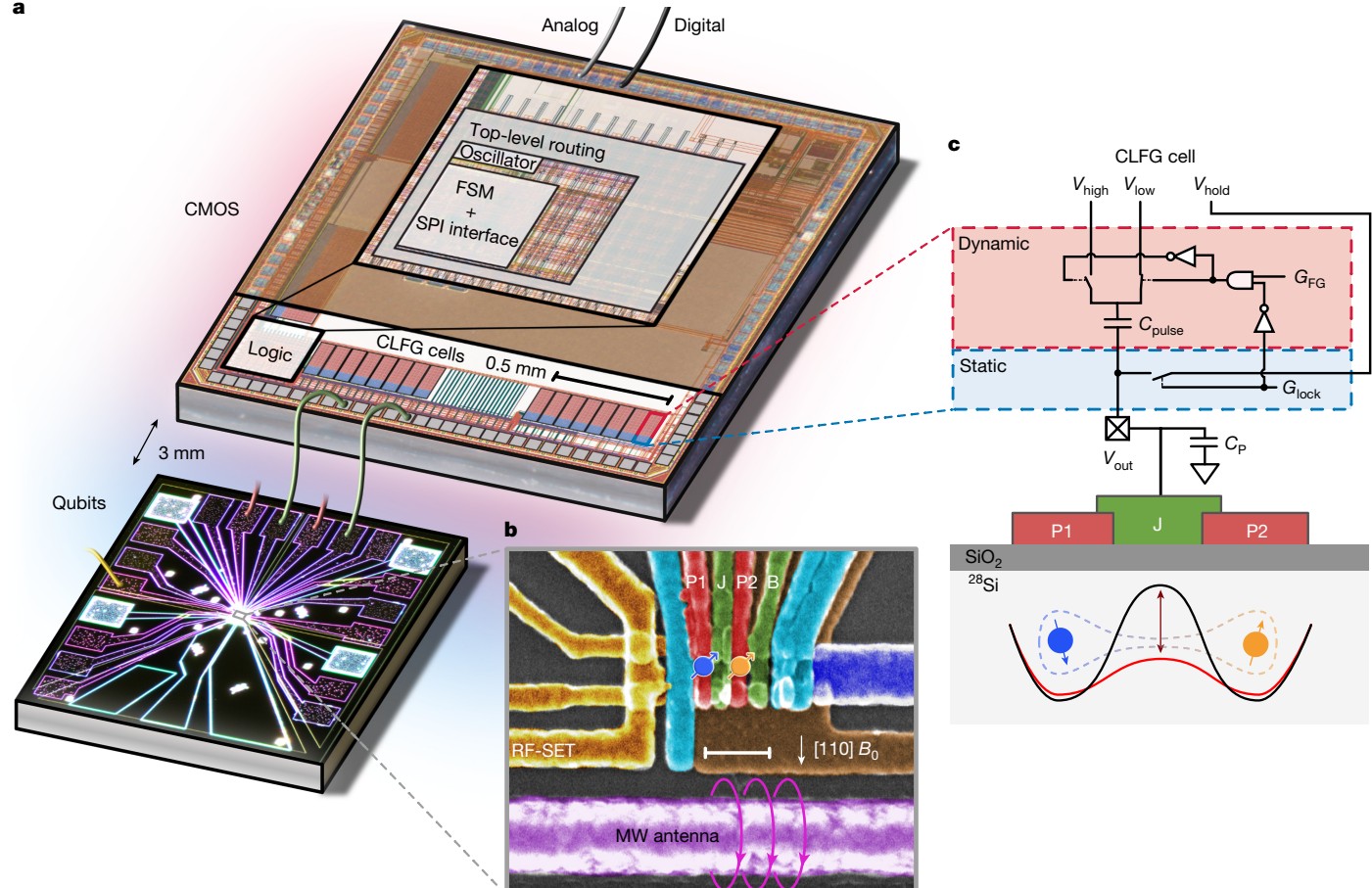

**Fig. 1 | Device and basic CMOS operation. a,** The cryo-CMOS and the qubit chip are mounted on the same circuit board at milli-kelvin temperatures and wire-bonded together. All other control systems are at room temperature. **b,** An electron micrograph of a nominally identical silicon device to that measured here. There are 32 CLFG cells on the chip, one of which is connected to gate J that controls the coupling between the two quantum dots on the qubit chip. The other cell is connected to gate B, which acts as an additional barrier gate. All other fast-pulse gates (P1, P2 and SET) and the d.c. barrier gates are connected to room-temperature electronics. **c,** Schematic of a cell and its electrical connection to gate J of the silicon device. Pulsing on this gate acts to modulate the tunnel coupling between the two quantum dots needed for single- and two-qubit controls. Scale bar, 100 nm (**b**).

chiplet architecture, comprising some 100,000 transistors, does not lead to a measurable reduction in coherence time.

## Experimental platform

An electron micrograph of a silicon-metal-oxide-semiconductor qubit device is shown in Fig. 1b. The device is fabricated on an isotopically purified $^{28}$Si epilayer with a residual $^{29}$Si concentration of 800 ppm (ref. 21) and $SiO_2$ isolating layer with metal gates patterned in aluminium. Quantum dots hosting single spin qubits are formed under the plunger gates (P1 and P2) at the $Si/SiO_2$ interface, and an exchange gate (J) modulates the tunnel coupling between the two dots, essential for two-qubit operations. A radiofrequency single-electron transistor (RF-SET)[22] detects the charge state of the quantum dots on microsecond timescales by leveraging an off-chip LC resonator operating near 400 MHz (ref. 23), and a proximal microwave antenna generates an oscillating magnetic field for spin resonance control (Fig. 1b).

The exchange gate J and a barrier gate B are wire-bonded to the cryo-CMOS control chip[7], which is implemented in 28 nm fully depleted silicon-on-insulator technology (FDSOI). The chip contains a serial peripheral interface for handling digital input instructions and a finite state machine (FSM) for on-chip digital logic. The FSM configures 32 analogue 'charge-lock fast-gate' (CLFG) circuit blocks, each of which can be used to control a gate electrode on the quantum device (Fig. 1c). In this configuration, the charge is periodically stored and shuffled between small capacitors, leveraging the low leakage of transistors at cryogenic temperatures that maintain the potential during quantum operations. The cryo-CMOS chip also incorporates a ring oscillator and configurable register designed as a programmable internal trigger (see Extended Data Fig. 3a for oscillator schematics). Here, for convenience, we opt for external triggering.

The core functionality of a CLFG cell is to lock a static voltage bias and enable a fast pulse between two voltage levels, as outlined in Fig. 1c. For instance, targeting gate J, the CLFG cell is programmed to first bring the gate to a potential $V_{out}$, equal to the potential $V_{hold}$ of an external source. Opening the switch $G_{lock}$ under the control of the FSM 'charge locks' this potential on the gate capacitor. Although this floating capacitor is now galvanically disconnected from the source, a pulse can be induced by toggling the potential on the top plate of this capacitance between $V_{high}$ and $V_{low}$, as shown in Fig. 1c. This toggling is produced autonomously by the programmed on-chip FSM, leading to a modified output $V_{out}$ by $\Delta V_{pulse} = (C_{pulse}/C_P + C_{pulse}) \times (V_{high} - V_{low})$, where $C_P$ is the parasitic capacitance. This mechanism has previously been shown to produce pulse amplitudes of 100 mV at a power of about 20 nW MHz$^{-1}$ (ref. 7). Below, we demonstrate how this architecture can be used to efficiently control spin qubits.

## Experimental results and demonstrations

To evaluate single-qubit gates, we first establish a baseline using all room-temperature electronics for control. Following the usual protocol

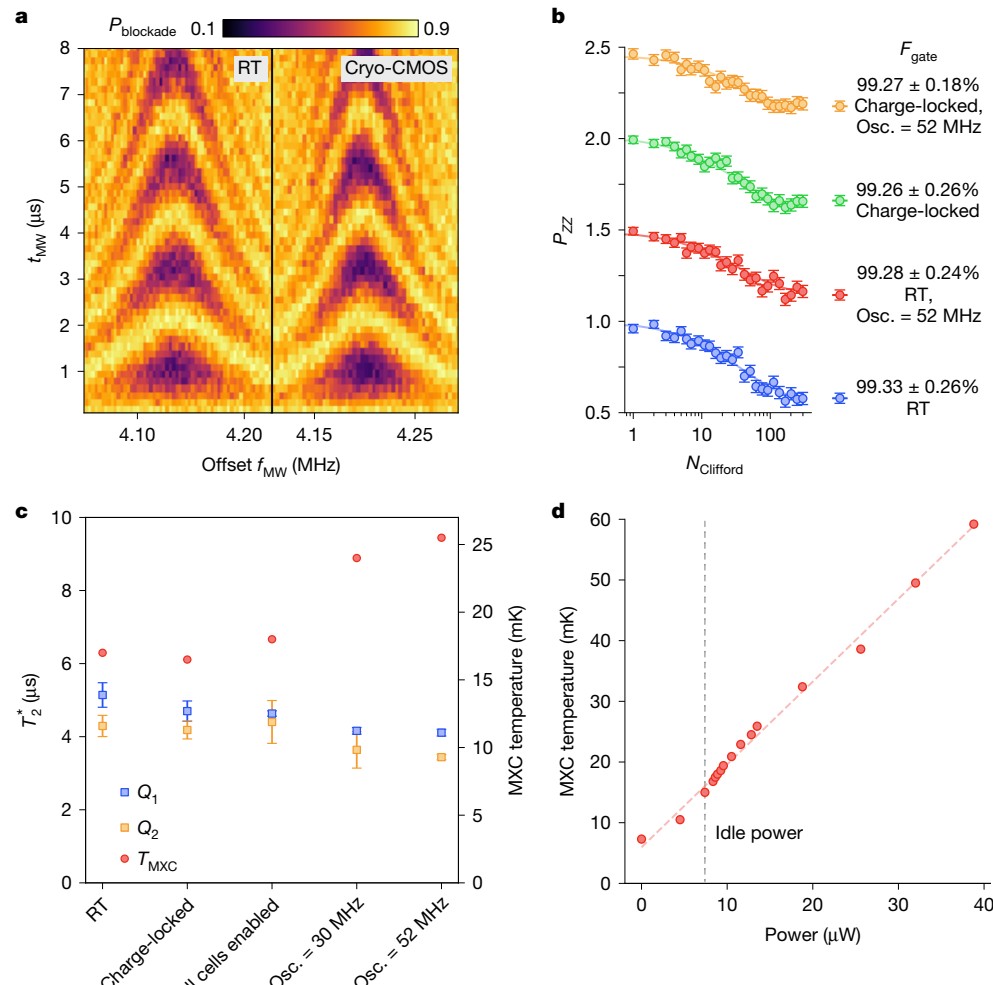

**Fig. 2 | Benchmarking single-qubit cryo-CMOS performance. a**, Single-qubit Rabi oscillations ($Q_1$) as a function of microwave (MW) frequency $f_{MW}$ and pulse time $t_{MW}$, performed with room temperature (RT) control. **b**, Single-qubit randomized benchmarking ($Q_1$) under various cryo-CMOS conditions. Traces are offset for clarity. Each data point is the average of 300 randomized sequences at 100 shots each. **c**, Single-qubit $T_2^*$ coherence time as a function of select cryo-CMOS parameters. Unless otherwise indicated, all data use cryo-CMOS control. Each data point is the average of 100 shots with 4 repeats for a total of 400 single shots. **d**, Mixing chamber temperature with cryo-CMOS power. Error bars represent the 95% confidence level. Osc., oscillator.

for two-spin manipulation[1,24,25], the singlet state is first prepared in the (1, 3) charge configuration using pulses applied to detuning gates P1 and P2, with ($n$, $m$) labelling the number of electrons in each dot under P1 and P2, respectively. A pulse applied to the J gate, connected to $V_{hold}$, then increases the barrier, separating the two electrons into each dot, in which they are independently addressed using the microwave antenna through their unique resonance frequency ($f_{ESR}$ = 13.9 GHz for a field $B_0$ = 0.5 T). Free-induction decay (FID) of the target spin is produced by applying microwave power to the on-chip electron spin resonance (ESR) line. Finally, a second pulse of the J gate returns the spins to the readout configuration, in which Pauli spin blockade enables spin-to-charge conversion[26,27] and measurement by the RF-SET. The shot-averaged readout signal as a function of microwave pulse time and frequency is shown in Fig. 2a. Beyond FID, we further establish our room temperature baseline by performing pulse sequences implementing Hahn echo (to measure coherence time $T_2$; Extended Data Fig. 4) and randomized benchmarking (to measure qubit control fidelity)[28,29]. We have intentionally limited the number of qubit gates bonded to the CMOS control chip to facilitate direct comparison with room temperature control in the same cooldown. The present prototype CMOS chip contains 32 CLFG cells to drive 32 qubit gate electrodes.

In this single-qubit measurement, the function of the J-gate pulse is to separate the two-spin system for controlled rotation by spin resonance.

As such, electrical noise, coupled through the J gate or other means, is unlikely to affect qubit fidelity in the limit that the pulse amplitude and duration are sufficiently large to fully separate the spins. Even so, we now evaluate the impact of cryo-CMOS control on single-qubit performance by performing the same protocol outlined above, but now with charge-locking applied to the J gate and the pulse produced using a CLFG cell under control of the FSM. Again, we generate FID data and quantitatively compare the CMOS and room temperature control using randomized benchmarking protocols, as shown in Fig. 2b. A slight degradation in qubit fidelity is observed (0.07%), probably because of unmitigated heat from the CMOS. We discuss heating in detail below.

Although electrical noise at the J gate does not directly couple with single spins, heat and drift in gate potential over longer timescales can affect qubit performance. Gate noise can also produce d.c. Stark shift of the qubit frequency in certain regimes (discussed further below). To investigate these mechanisms, we extract the time-ensemble average coherence time $T_2^*$ for each qubit, repeatedly measured as each circuit block of the CMOS chip is powered up. Comparing the data in Fig. 2c again shows a small impact with respect to our room temperature baseline, correlating with a slight rise in the base temperature of the refrigerator (see Fig. 2d and its caption for details).

A drawback of the control scheme outlined above is its reliance on qubit-specific microwave pulses, which, despite cryo-CMOS gate

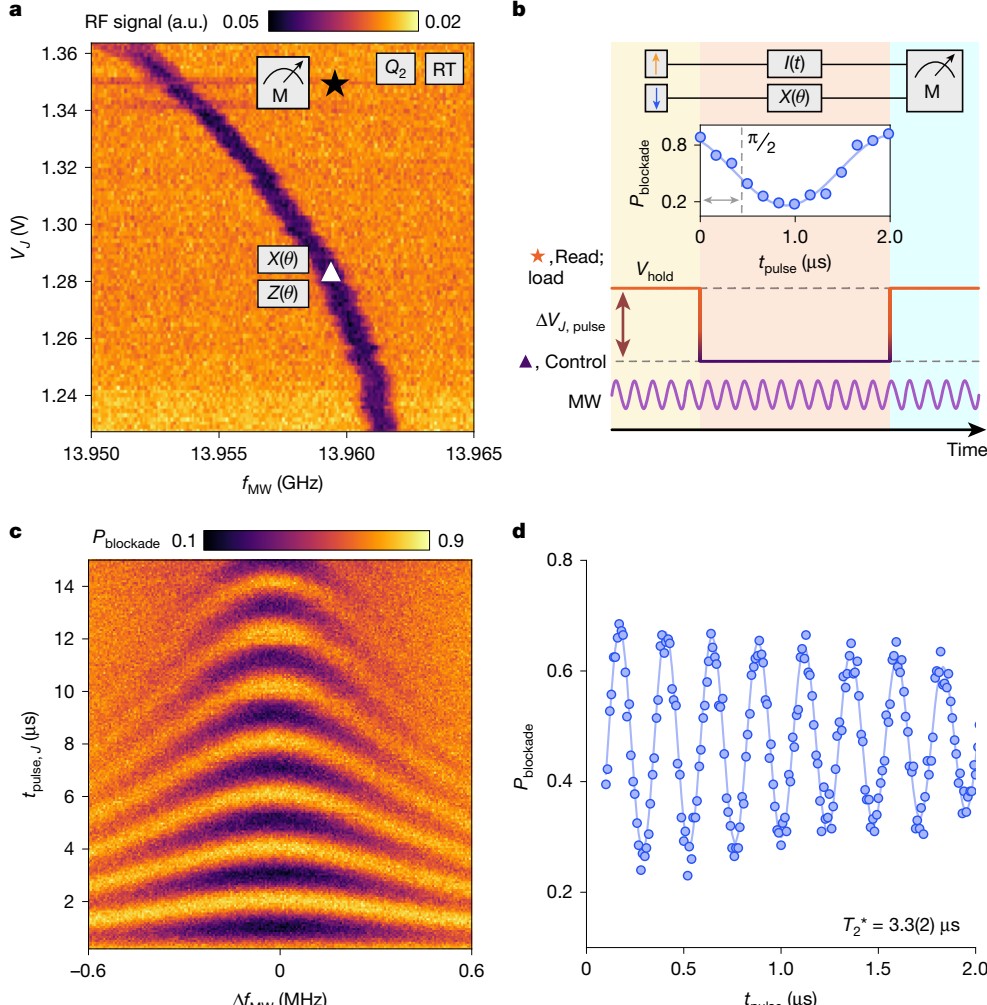

**Fig. 3 | Single-qubit gates with cryo-CMOS. a**, ESR spectra as a function of microwave (MW) frequency and $J$ gate voltage. $Q_2$ exhibits a significant Stark shift, which allows for on-resonance single-qubit operations ($X$, $Z$; triangle), and off-resonance loading and measurement (M; star). **b**, Schematic of the pulsing sequence when using effective global control. Qubit rotations are determined by $J$ pulse time, with the microwave tone fixed to the maximum $t_{pulse}$ time, extending into load and read stages. **c**, Rabi chevron of $Q_2$. **d**, Ramsey coherence time, using global control and cryo-CMOS pulsing at $B_0 = 0.5$ T. Error margin represents the 95% confidence level. a.u., arbitrary units.

control, require additional room-temperature microwave generators (and cables) for each qubit. We next demonstrate an alternate control approach that leverages a continuous wave global microwave field, sourced from room temperature but common to all qubits[15,30]. Key to this scheme is the ability to tune the spin resonance frequency of a qubit using a gate voltage[31,32]. This d.c. Stark shift enables a gate pulse to bring a qubit into resonance with the global field for a controlled amount of time to produce a rotation in the qubit state vector. With a single microwave tone from room temperature, cryo-CMOS produces the 'baseband' gate pulses that independently bring each qubit into and out of resonance with the global field.

The global microwave scheme requires a calibration of the unique d.c.-Stark shift produced by the J gate on, for example, qubit $Q_2$, as shown in Fig. 3a. The sizable shift (about 1 MHz per 10 mV) is well-matched to the voltage pulses that can be efficiently generated with proximal low-power CMOS. Here, the spins are initialized to a mixed or $T^-$ ($|\downarrow\downarrow\rangle$) state off-resonance by a detuning pulse to a relaxation hotspot[33,34]. The J-gate pulse produces the time-controlled Stark shift as shown in Fig. 3b. Repeating this sequence as a function of pulse length yields the coherent oscillations (Fig. 3c), and coherence metrics can be extracted (Fig. 3d). For this measurement, the width of the pulse is set by the timing of the trigger fed to the CMOS from room temperature.

Conceptually, this reliance on room temperature triggering may seem to be a limitation of our CMOS circuits. However, we note that fine time resolution is needed only to map out coherent oscillations. By contrast, once calibrated, logic gates require a fixed time pulse, for instance, 1.034 µs to produce a π/2 rotation. As such, these fixed time pulses are straightforward to implement with our CMOS architecture. Furthermore, we note that high fidelity control is possible with fixed time width pulses by precise tuning of the pulse amplitude and bias, shifting the requirement for high-resolution timing to the resolution of the voltage source.

Finally, we turn to evaluate two-qubit logic gates, which provide the most stringent test for crosstalk or electrical noise stemming from proximal milli-kelvin CMOS control. Here, the target qubit is rotated about the $z$-axis depending on the state of the control qubit, with the gate-tunable exchange interaction modulating the coupling between the two electrons[1,24]. As a baseline, we first perform a decoupled controlled phase gate (DCZ) using all room temperature instrumentation. The DCZ gate incorporates a spin-echo sequence with coherent rotation about the $z$-axis of angle $\phi = J(\epsilon)t_{exchange}\hbar$, enabled by turning on exchange for a controlled time with J-gate pulse of duration $t_{exchange}$ (Extended Data Fig. 2d). The resulting readout probability with J-gate pulse width is shown in Fig. 4a. The DCZ gate is sensitive to high-frequency electrical noise arising either directly from exchange-gate voltage fluctuations

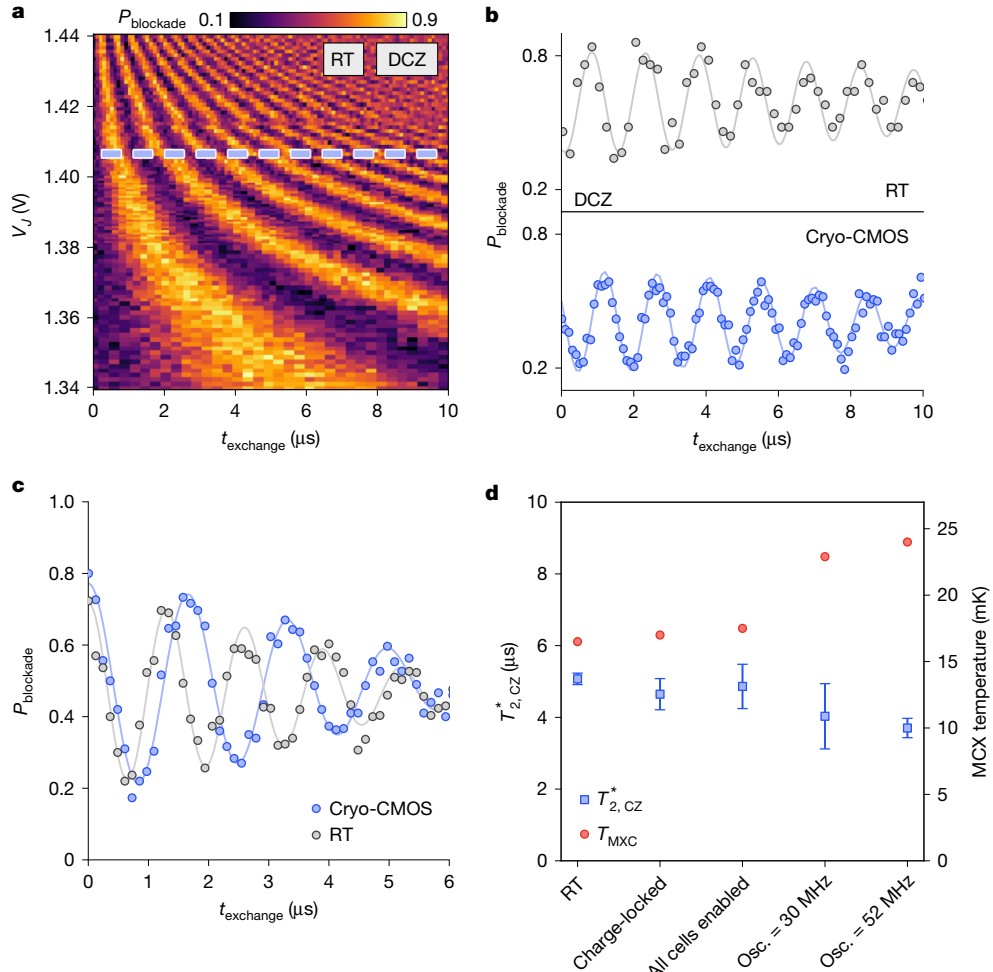

**Fig. 4 | Two-qubit operations using cryo-CMOS. a**, DCZ oscillations as a function of exchange time $t_{exchange}$ and $V_J$, performed using room-temperature (RT) control. **b**, Room-temperature- and cryo-CMOS-enabled DCZ oscillations at a set $V_J$, indicated in **a**. Set level is determined by $V_{hold}$, which can be tuned for stronger qubit interaction. **c**, Comparison of visibility between room-temperature and cryo-CMOS control using identical pulsing methods on all gates. Increased state preparation and measurement error is present because of two-level-only approach to pulsing $V_J$. **d**, CZ coherence times of our two-qubit control conditions under various cryo-CMOS parameters. $J$ pulses are generated by cryo-CMOS unless otherwise indicated. Each data point is the average of 100 shots, with 4 repeats for a total of 400 single shots. Error bars represent the 95% confidence level. Osc., oscillator.

or indirectly from noise in the (gradient) magnetic field or variation in $g$-factors from gate-induced movement in the position of the electron wavefunction.

Comparing cryo-CMOS control with our room-temperature baseline, we observe that the coherent exchange oscillations show similar behaviour (Fig. 4b). These results immediately confirm the utility of proximal milli-kelvin CMOS for controlling two-qubit logic gates. Close inspection perhaps suggests a suppression in visibility for the CMOS data, which probably stems from the limited two-state resolution of the voltage pulses used to tune the readout and preparation state, which for this qubit device require significantly different tunnel rates than those used in two-qubit control. Although it is not uncommon to find tunnel rates that are very similar for qubit control as state preparation and measurement (SPAM), in the present device, the differing tunnel rates precluded more quantitative measures of SPAM error and two-qubit fidelity using cryo-CMOS[35]. Future improvements in qubit tunability and fidelity will also enhance sensitivity to new noise sources, including further assessment of the control platform.

As a noise diagnostic tool, it is also worth noting that the DCZ gate is limited because the spin-echo sequence decouples the spin dynamics from low-frequency noise. Removing the echo pulses then opens the bandwidth to now include all of the low-frequency components down to quasi-d.c., offering a better measure of the total aggregate noise inherent in the system. A comparison of room-temperature control and cryo-CMOS is made in the data shown in Fig. 4c, now without spin-echo. These datasets constitute a measure of the ensemble average coherence time associated with the exchange gate, $T^*_{2,CZ}$. Finally, using this parameter as a wideband measure of noise, Fig. 4d compares $T^*_{2,CZ}$ for room-temperature control, cryo-CMOS with a single charge-locked cell, all 32 cells locked (mirroring $J$-gate pulses), and as a function of CMOS oscillator frequency. A slight reduction in $T^*_{2,CZ}$ (around 20%) is observed at the highest clock frequencies, which, given the slight corresponding increase in refrigerator temperature, can be explained as arising from parasitic heating (for more detailed two-qubit performance data see Extended Data Fig. 5). We note that no additional (electrical) noise is observed beyond the thermal noise contribution associated with the small increase in temperature from CMOS power dissipation (for further discussion, see Extended Data Fig. 4).

## Discussion

Our cryo-CMOS control chip consists of complex mixed-signal circuits realized using more than 100,000 transistors. Most of these transistors are used in the digital sub-systems and related circuit blocks,

accounting for a fixed overhead power of tens of microwatts. On top of this constant offset power from the digital blocks, the CLFG analogue cells each contribute approximately 20 nW MHz$^{-1}$ when generating 100 mV amplitude pulses, enabling many thousands of cells (and thus gate pulses) to fit within the cooling budget of a commercial dilution refrigerator (around 1 mW at 100 mK) (ref. 7). Apart from the cooling limits of the refrigerator, however, a challenge arises in the thermal management of hot control systems to ensure the routing of heat bypasses proximal, cold quantum devices. Here, we have made no attempt to mitigate this parasitic heating, simply wire-bonding the chips together in a standard package. This arrangement can lead to elevated electron temperatures in the quantum device (Extended Data Fig. 3) even when the refrigerator remains cold (Fig. 2d) and is the likely explanation for the small impact we observe in qubit fidelity when the largest CMOS circuits are powered up at the highest clock rates. As such, we emphasize that there is a notable opportunity to suppress parasitic heating by using separate parallel cooling pathways for the CMOS chip and quantum plane[7]. The use of heterogeneous, rather than monolithic, integration opens new thermal configuration options in this regard.

Beyond direct heating, the close presence of 100,000 transistors, with volt-scale biasing and sub-nanosecond rise and fall times, can create an exceedingly noisy environment in which to operate electrically sensitive qubits. It is surprising that the CMOS chip has only a small impact on qubit performance relative to previous experiments with room-temperature control[17,19]. Furthermore, the small degradation in fidelity is probably explained entirely from parasitic heating, rather than from electrical noise. Certainly, our use of CMOS design rules that minimize external crosstalk are important; however, beyond these, we suggest three additional aspects that probably reduce electrical noise. First, as the physical temperature of the CMOS die is a few hundred milli-kelvin, thermal noise contributions are substantially suppressed. Second, the chip-to-chip interconnect probably has a relatively low bandwidth, filtering noise above a few gigahertz. Last, we note that the action of the CLFG circuits effectively decouples the CMOS from the quantum device when in charge-lock mode, except for a very small coupling capacitor. Taken collectively, these aspects further underscore the utility of heterogeneous over monolithic integration for mitigating crosstalk and heating. Apart from addressing the challenges posed by scaling up qubits, cryo-CMOS using a chiplet architecture may also prove useful in generating ultrafast, low thermal noise control pulses that probe fundamental physics in mesoscale quantum devices[36].

In conclusion, the results presented here demonstrate the viability of heterogeneous, milli-kelvin CMOS for generating the volt-scale biases and milli-volt pulses needed to control spin qubits at scale. Beyond addressing the interconnect bottleneck posed by cryogenic qubit platforms, these results show that degradation in qubit performance from milli-kelvin CMOS is very limited. Although our focus here has been controlling spin qubits based on single electrons, we draw attention to the inherent compatibility of our control architecture with other flavours of spin qubits, for instance, exchange-only qubits that leverage square voltage pulses exclusively[37]. Pairing cryo-CMOS-based control with highly compatible radiofrequency readout approaches that exploit dense frequency multiplexing[38] enables a highly integrated and scalable spin qubit platform.

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

## Methods

### Measurement setup

Measurements are performed in a Bluefors LD400 dilution refrigerator with a base temperature of 7 mK. The qubit chip and cryo-CMOS chip are packaged on the same FR4 printed circuit board (PCB)[39], separated by 3 mm and wire-bonded together. The daughterboard PCB is placed on a custom motherboard and electrically connected by an interposer. The motherboard manages signals from room temperature, and the ESR line connects directly to the daughterboard through a miniSMP. The PCB setup is mounted in a magnetic field, on a cold finger. The external magnetic field is supplied by an Oxford MercuryiPS-M magnet power supply. The d.c. voltages are generated by an in-house custom-made digital-to-analogue converter (DAC). A Quantum Machines OPX+ generates room-temperature dynamic pulses, the 400 MHz signal for radiofrequency readout, I/Q and pulse modulation for the microwave source as well as trigger lines to the cryo-CMOS chip and microwave source (Extended Data Fig. 1). Dynamic and d.c. voltage sources are combined at room temperature using custom-made voltage combiners. The OPX has a sampling rate of 1 GS s$^{-1}$ and a clock rate of 250 MHz. The microwave tone is generated by a Keysight PSG E8267D Vector Signal Generator with a signal spanning 250 kHz to 31.8 GHz. Single-qubit gates are operated at a microwave frequency of 13.9 GHz.

We use an RF-SET and radiofrequency reflectometry readout, comprising a Low Noise Factory LNF-LNC0.3-14A amplifier at the 4 K stage of the refrigerator and a Minicircuits ZX60-P103LN+ at room temperature for signal amplification. The directional coupler in Extended Data Fig. 1 is a Minicircuits ZEDC-10-182-S+ 10–1,800 MHz.

All measurements are performed in the same cooldown to enable careful comparison between control methods without device variation that can occur from thermal cycling. For room-temperature operation, a pass-through switch $G_{lock}$ is closed and room-temperature-sourced exchange gate pulses are delivered through $V_{hold}$ (see Extended Data Fig. 1 for instrument connection details).

### Cryo-CMOS programming

Our cryo-CMOS receives programming instructions, power, d.c. bias and dynamic voltage levels from an in-house room-temperature DAC[40] and is configured to receive an external trigger from the OPX+, as well as room-temperature dynamic pulses for pass-through room-temperature control. Some of the programming instructions as well as the external trigger, are received by the same input line. Appropriate input is handled by a Minicircuits RC-4SPDT-A18 DC-18 GHz radiofrequency switch, which takes inputs from both the DAC and OPX+. The radiofrequency switch is programmed to work in unison with charge-locking commands sent to the cryo-CMOS, switching inputs from the DAC to the OPX+ once programming is complete.

Owing to the low leakage rate of our cryo-CMOS transistors, there are no charge-lock refresh cycles of our charge-locked gate when performing experiments. Maximum experiment times are approximately 1 h, usually single-qubit RBM or PSD measurements. Cryo-CMOS circuit blocks were measured to be functional within the bias range DD1P0 = 0.65V, N/PMOS backgate = 1.5/−1.5 V, VDD1p8 = 1.2 V−1.6 V.

## Data availability

The datasets generated and/or analysed relevant to this study are available from the corresponding author upon request and further available at Zenodo (https://zenodo.org/records/15080656)[41].

## Code availability

Analysis codes supporting the findings of this study are available from the corresponding authors upon request and further available at Zenodo (https://zenodo.org/records/15080656)[41].

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

**Acknowledgements** We thank Y. Yang and R. Kalra for technical contributions and discussions. This research was supported by Microsoft Corporation (CPD 1-4) and by the Australian Research Council Centre of Excellence for Engineered Quantum Systems (EQUS, CE170100009). We acknowledge support from the Australian Research Council (FL190100167 and CE170100012), the US Army Research Office (W911NF-23-10092), the US Air Force Office of Scientific Research (FA2386-22-1-4070) and the NSW Node of the Australian National Fabrication Facility as well as the Research and Prototype Foundry Core Research Facility at the University of Sydney, also part of the Australian National Fabrication Facility. The views and conclusions in this document are those of the authors and should not be interpreted as representing the official policies, either expressed or implied, of Microsoft Corporation, the Army Research Office, the US Air Force or the US government. The US government is authorized to reproduce and distribute reprints for government purposes, notwithstanding any copyright notation herein. R.Y.S. and S.S. acknowledge support from the Sydney Quantum Academy.

**Author contributions** S.K.B., W.G., K.Z., T.T., C.H.Y., N.D.S., R.Y.S., S.J.P. and D.J.R. designed the experiments. K.D., S.J.P. and D.J.R. designed and characterized the CMOS chip. S.K.B. and K.Z. performed experiments with input from W.G., K.D., T.T., S.S., C.C.E. and A.L., under the supervision of A.S.D. and D.J.R.; W.H.L. and F.E.H. fabricated the device on enriched $^{28}$Si wafers supplied by K.M.I.; S.K.B., K.Z. and D.J.R. wrote the paper with input from all co-authors.

**Funding** Open access funding provided by the University of Sydney.

**Competing interests** A.S.D. is the CEO and director of Diraq. S.K.B., W.G., T.T., C.H.Y., N.D.S., W.H.L., C.C.E., F.E.H. and A.L. declare equity interest in Diraq. D.J.R. is the CEO and Director of Emergence Quantum. K.D. and S.J.P. declare equity interests in Emergence Quantum. Other authors declare no competing interests.

**Additional information**
**Correspondence and requests for materials** should be addressed to David J. Reilly.

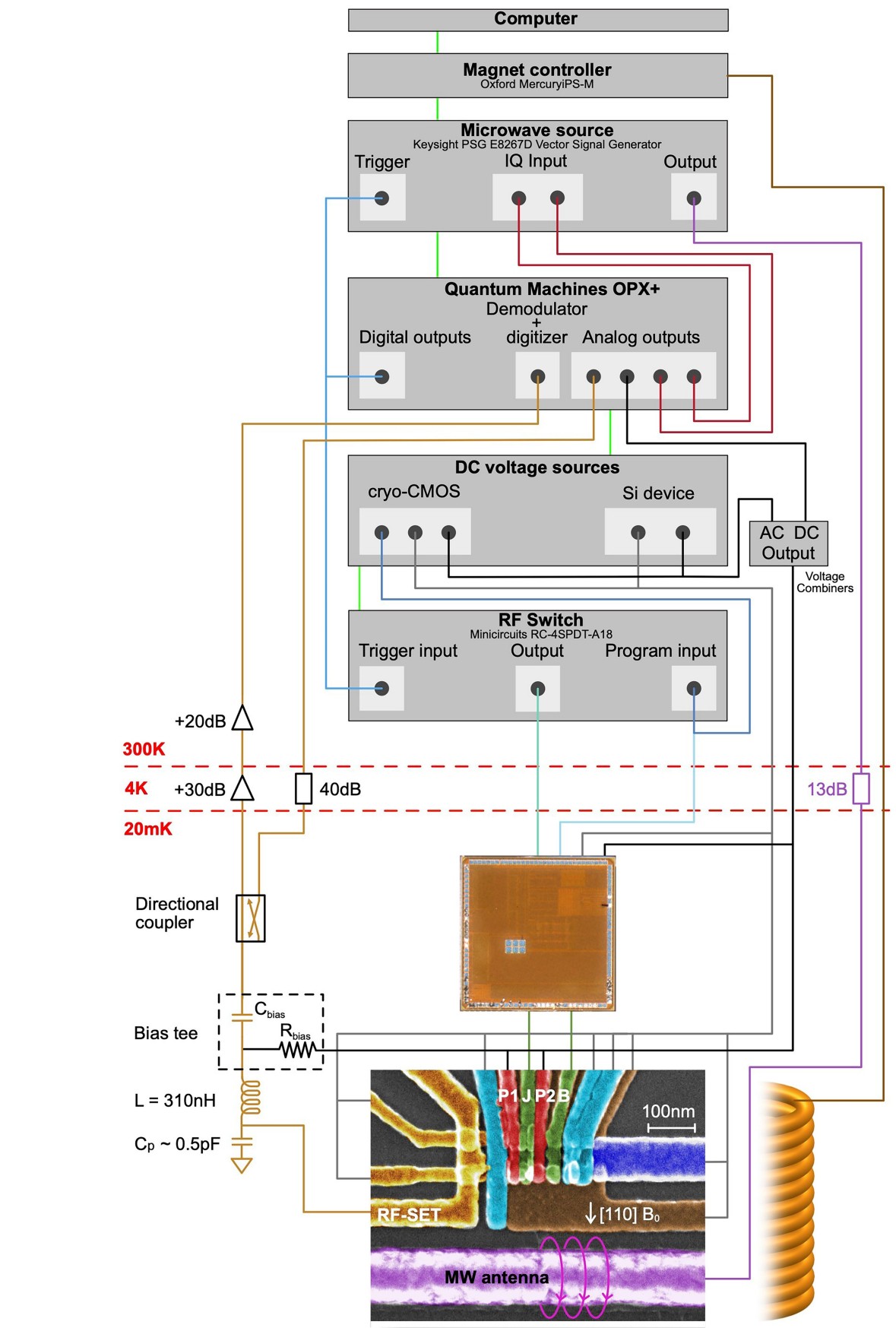

**Extended Data Fig. 1 | Full experimental setup schematic.** For further hardware information, see Methods.

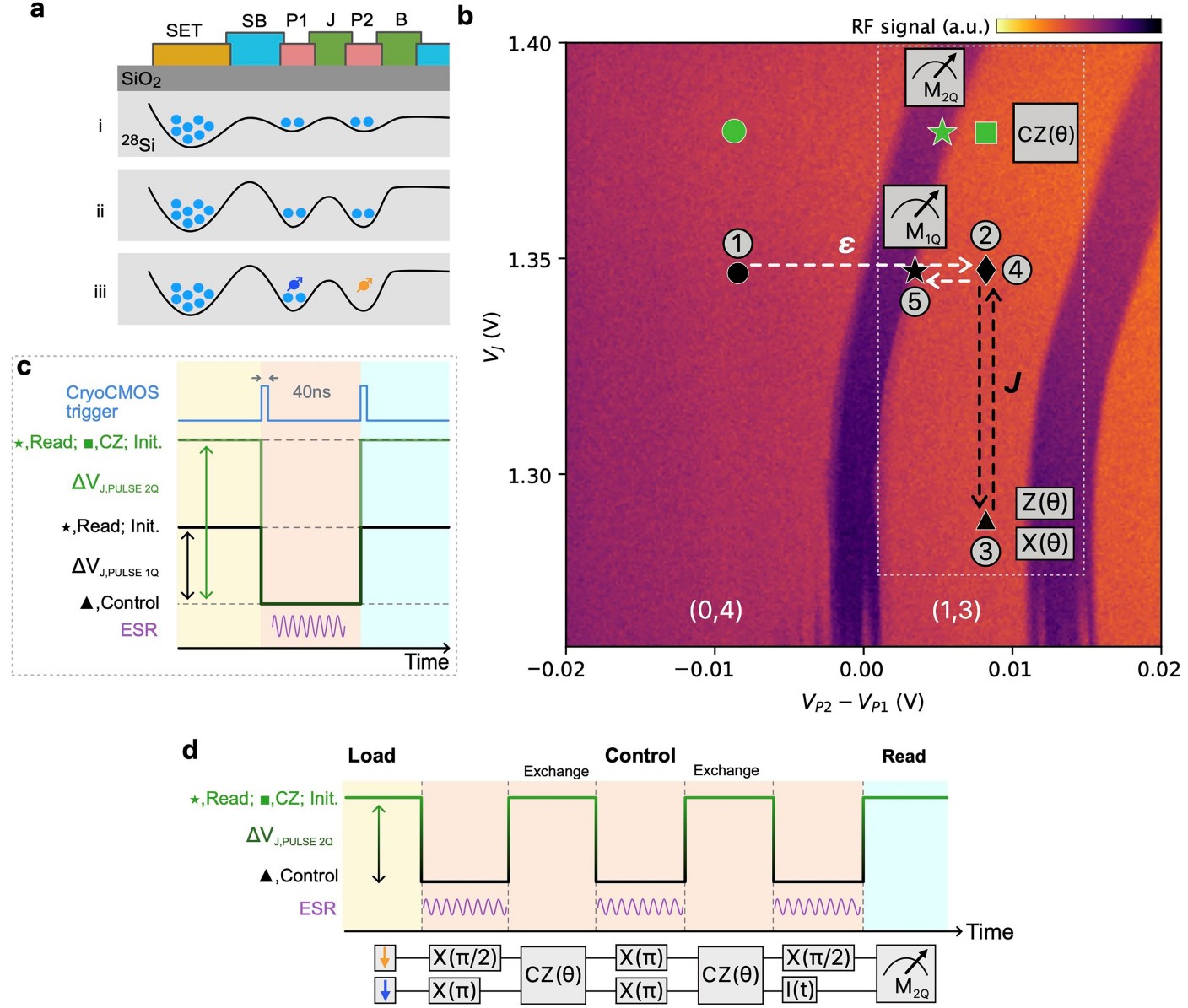

**Extended Data Fig. 2 | Loading and pulsing scheme. a**, the barrier to the SET reservoir is lowered for a set number of electrons to occupy the dots (i). This barrier is raised to isolate the dots (ii), and the electron states are initialized via tuning of the plunger gates P1 and P2 (iii). **b**, Single-qubit pulsing scheme for ESR measurements. A diabatic detuning ramp (1→2) from the (0,4) to (1,3) state initialises the double quantum dot into a singlet state. The qubit is then pulsed using the *J*-gate to the control point (2→3) at which point a MW pulse is applied that rotates the target qubit in resonance with the MW frequency. The qubit is pulsed back to ◆ (3→4), then pulsed into the Pauli Spin Blockade regime (4→5) for readout. Detuning ε pulses use gates *P*1 and *P*2 and are always RT-operated. *J*-gate pulses are generated either at RT or by cryo-CMOS. The approximate operation points of single qubit (X, Z), two qubit (CZ), and readout points ($M_{1Q}$, $M_{2Q}$) are indicated by a triangle (▲), square (■), star (★) and diamond (◆) respectively. **c** (inset of b), basic *J*-gate pulsing scheme for ESR measurements when using cryo-CMOS. Here, the gate *J* is in a "charge-locked" state held at $V_{HOLD}$. An external trigger is used to pulse between the two *J*-gate levels, whose separation is determined by $\Delta V_{J,PULSE}$. **d**, DCZ sequence focusing on *J*-gate pulses. The qubits are initialised into a $T_-$ state, before *X* and *X*/2 gates are performed on the control and target qubits respectively. A spin-echo sequence is inserted between the exchange gates, which precedes the final projection for measurement of the two-qubit spin state.

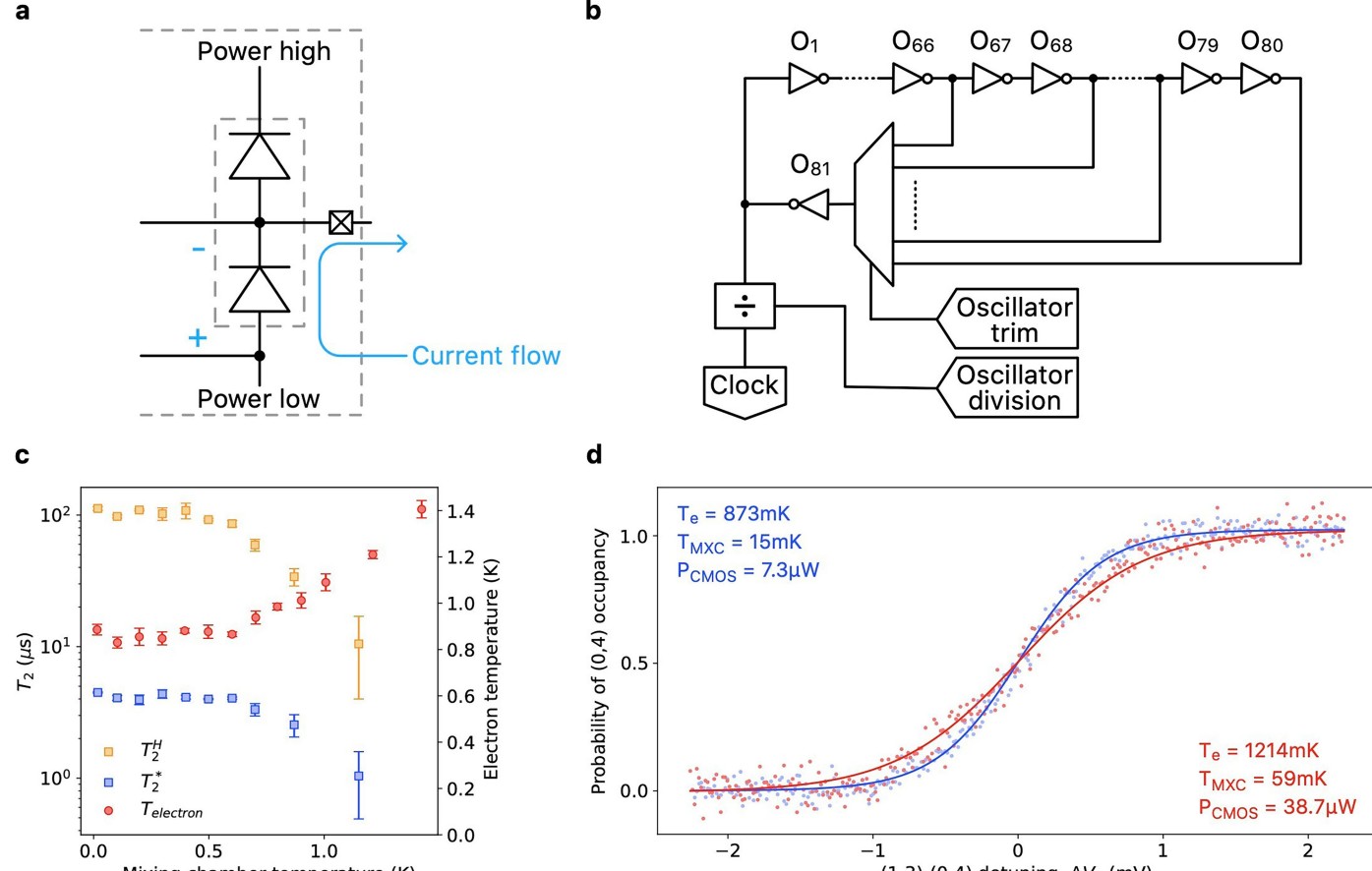

**Extended Data Fig. 3 | Temperature dependence using artificial cryo-CMOS and mixing chamber heating. a**, By forward biasing an ESD protection diode on-chip, the power draw can be programmed to mimic that of any oscillator trim or division value upon oscillator activation. Artificial power draw also allows for higher resolution investigation versus other parameters, as well as extrapolation beyond what the maximal cryo-CMOS power draw possible with it's feature set. **b**, Schematic of the cryo-CMOS ring oscillator. 81 in-series inverters are connected to a three-bit multiplexer, controlling the tap-off point. The frequency is tunable by programmable inputs into the oscillator trim. This frequency is further divided by the oscillator divider, which is eight-bit tunable. The ultimate output frequency is then passed on to the FSM.

**c**, Hahn echo coherence time and Ramsey coherence time of Q1, and measured electron temperature as a function of mixing chamber temperature. Base effective electron temperature is approximately 850 mK, which only starts to increase once the mixing chamber exceeds 700 mK. Electron temperature and mixing chamber temperature equalize at 1 K. Each data point consists of 100 shots with 4 repeats for a total of 400 single shots. Error bars represent the 95 % confidence level. **d** (1-3)-(0,4) charge occupation probability. Solid lines are Fermi distribution fits, allowing for extraction of effective electron temperature. Fitting to distributions with high mixing chamber temperature (at which point becoming equal to that of the sample) allows for determination of the lever arm.

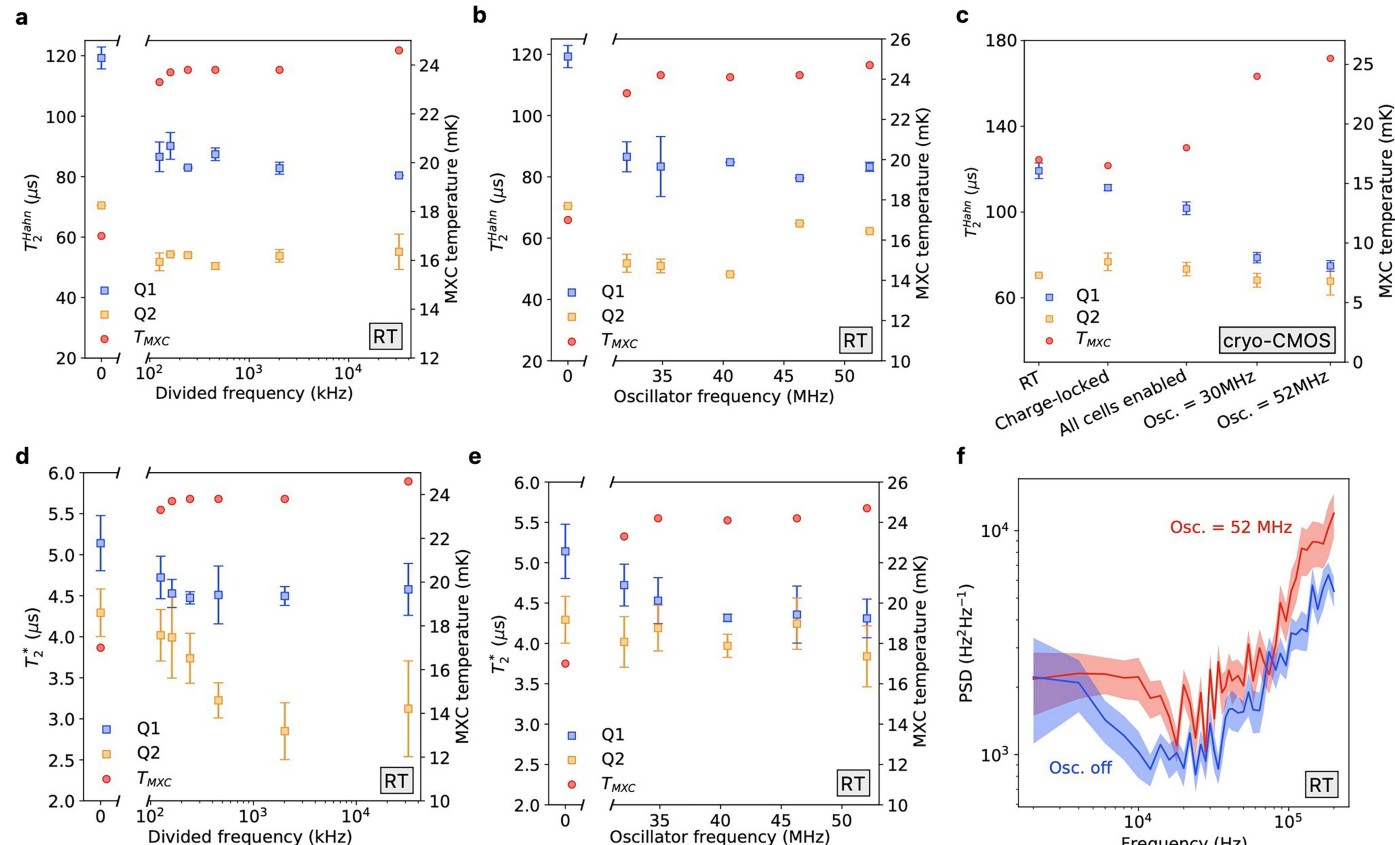

**Extended Data Fig. 4 | Extended single qubit performance. a**, $T_2^{Hahn}$ coherence of both qubits as a function of divided frequency from the cryo-CMOS oscillator. The output clock frequency (here set to 30 MHz) passed to the fast state machine is divided by an integer between 1 and 255, see Extended Data Fig. 2 for schematic details. The inset indicates whether RT or cryo-CMOS control is used; all RT pulsing schemes are replicable by cryo-CMOS. Activating the oscillator immediately causes a drop in coherence due to the extra thermal dissipation, and lowering the divider value increases this dissipation slightly. **b**, Qubit coherence while directly changing oscillator frequency. The divider is set to a constant value of 255, and increasing oscillator frequency leads to an increase in power draw of the cryo-CMOS chip, reflected in the mixing chamber temperature. **c**, Similar to Fig. 2(c), $T_2^{Hahn}$ coherence is also explored as a function of various parameters under cryo-CMOS control conditions. **d**, **e**, $T_2^*$ of both

qubits while changing the divided and oscillator frequencies, similar to a and b. For all data points, each data point consists of 100 shots with 4 repeats for a total of 400 single shots. **f**, The noise power spectral density (PSD) is explored as a function of oscillator frequency. Noise spectroscopy, based on the Carr-Purcell-Meiboom-Gill (CPMG) protocol[42–44], uses a single qubit as a noise probe. A slight increase in the overall white noise level is observed over the detectable frequency range when the oscillator is at its maximum frequency. This increase in white noise is consistent with parasitic heating leading to an increase in temperature and thermal noise contribution to the cumulative noise background. We note that we do not observe additional electrical frequency spurs or artifacts beyond the thermal noise contribution. At higher frequencies, we observe an increase in PSD, likely due to high-power driving or mis-calibration of microwave pulses[19,45,46]. Error bars represent the 95 % confidence level.

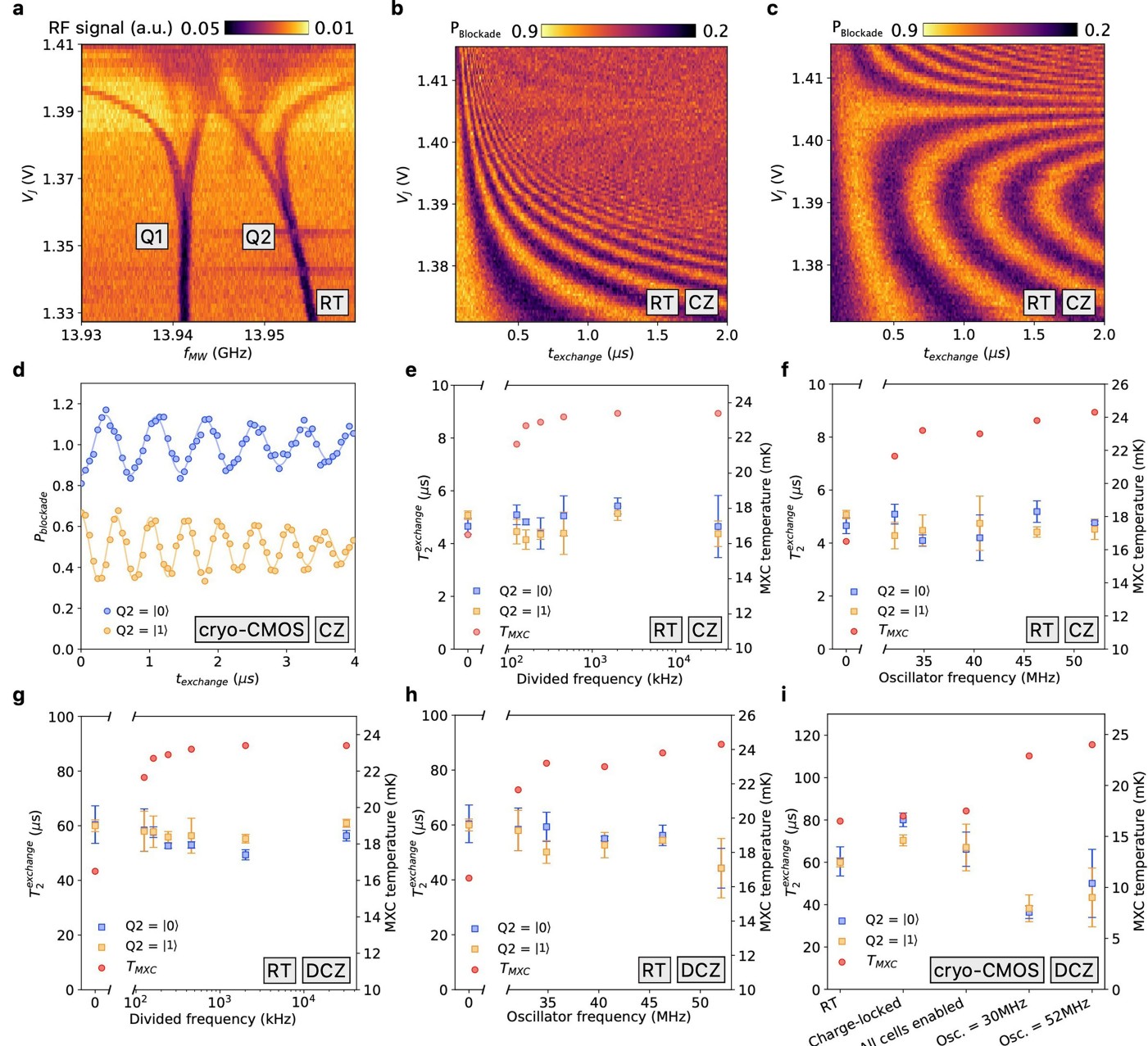

**Extended Data Fig. 5 | Extended two-qubit performance and dependence on cryo-CMOS parameters. a**, ESR spectrum as a function of $V_J$. Exchange starts to open at around $V_J = 1.37$ V. Measurements are done with room temperature $J$ pulses; RT or cryo-CMOS control is indicated in the lower right hand side of each figure. **b, c**, CZ oscillations as a function of exchange time $t_{exchange}$ and $V_J$. Multiple $J$ levels, not replicable by cryo-CMOS control are used here for optimal performance. The initial state of the target control qubits is indicated in the bottom right hand corner. In (b), the $J$ gate drives two-qubit exchange at a sensitivity of 25dec\$V$. **d**, CZ oscillations using cryo-CMOS control, showing the difference in visibility compared to **b-c** averaged over 100 shots. Traces are offset for clarity. For all further figures, pulses are two-level

and if generated at RT, are replicable by cryo-CMOS. **e**, similar to Extended Data Fig. 4(a), the two qubit CZ coherence time $T_2^{exchange}$ is explored as a function of divided frequency. The oscillator is set to 30 MHz for all divisions, and pulses are generated from room temperature. **f**, CZ coherence time $T_2^{exchange}$ is now explored as a function of oscillator frequency. The divider is set to 255 for all measurements. **g, h**, Measurements in **e, f** are replicated here, however now using a DCZ pulsing protocol, allowing for longer coherence times. **i**, DCZ coherence is explored using cryo-CMOS control under various parameters, similar to Fig. 4(d). For all individual data points, each data point consists of 100 shots with 4 repeats for a total of 400 single shots. Error bars represent the 95 % confidence level.