## [Peer Review File · Nature]

Spin-qubit control with a milli-kelvin CMOS chip

Corresponding Author: Professor David Reilly

Version 0:

Reviewer comments:

Referee #1

(Remarks to the Author)

This paper presents a demonstration of spin-qubit control by a millikelvin CMOS chip. It is astonishing that this control method works at millikelvin temperatures. Most previous work in this direction has used control chips at elevated temperatures, but the charge-locking mechanisms allows the chip to work at millikelvin temperatures. This is an important step for large-scale quantum computing.

My overall comment is that this paper would benefit from more precise arguments in support of the main claims. In my understanding, there are three main claims. First, the cryo-CMOS chip does not significantly reduce the qubit performance. Second, the cryo-CMOS chip does not introduce additional electrical noise (either through a fluctuating voltage or crosstalk), and third, the (slight) reduction in fidelity is likely due to excess heat and two-level-only pulse control.

About the first claim, the RB results support the claim that the single-qubit gate fidelities are not reduced. However, the measurements of state preparation, readout, and two-qubit gate performance are based on qualitative metrics, like visibility, and the decay of two-qubit oscillations. Why were SPAM and 2Q fidelities not measured? If these measurements are not easy to perform, and if the authors agree that these would be helpful, it would be good to state this in the conclusion.

About the second claim, certainly the fact that the qubit metrics do not diminish significantly compared to RT control is a strong argument. However, I wonder about the reduction in the metrics as more items are powered up. How do we know this is not due to electrical noise? Looking at the data of Extended Data Fig. 4f, for example, what is the mechanism behind the increase in the noise as the oscillator is turned on?

About the third claim, what exactly is the argument in support of this? I can see that TMC increases as more items are powered up, and I can imagine that this could increase, for example, the charge noise level experienced by the qubit as the qubit-chip temperature increases. But I could also imagine, for example, that the noise from the cryo-CMOS device also increases (see above). I get the impression that Extended Data Fig. 3 is supposed to help with this argument, but I can't quite make the link.

About the claim that two-level-only control increases SPAM errors, what is the argument in support of this? For example, do the authors have a model that shows if they include only two-level control, SPAM errors increase, and can they reproduce the observed data with the model?

Smaller comments include the following.

1. What is the uncertainty on the value of T_2^* in Fig. 3d? Is this T_2^* value meaningfully lower than the T_2^* values presented in Fig. 2c? If so, why?
2. Are the RT/cryo-CMOS comparisons done in the same cooldown? If so, how? Why are the frequencies in Fig. 4c different from each other?
3. The electron temperature seems quite high at 850 mK. Do the authors expect this is because of the introduction of the cryo-CMOS chip?
4. Are there prospects for more-than-two-level control?

Referee #2

(Remarks to the Author)

The manuscript titled "Spin Qubits with Scalable milli-kelvin CMOS Control" by Bartee et al. presents an interesting study in which the team utilizes ultra-low-power cryogenic CMOS electronics to control a Si MOS spin qubit processor. The low power consumption enables the controller to operate at milli-kelvin temperatures, in close proximity to the quantum chip, within a heterogeneous integration architecture. When operating the CMOS controller at the highest clock frequency generated by the ring oscillator, the mixing chamber plate temperature of a dilution refrigerator increases to approximately 25 mK, leading to a ~20% reduction in T_2^* and a ~50% reduction in T_2 echo.

The key finding of the manuscript is that single-qubit gate fidelity—and especially two-qubit CZ decoherence time—are minimally affected by the thermal and electrical noise emitted from the cryo-CMOS controller. The manuscript attributes the extremely low power consumption of the controller primarily to the charge-lock-fast-gate (CLFG) component, which exhibits negligible charge leakage at milli-kelvin temperatures, along with other unspecified improvements. Residual power consumption is mainly driven by transistor switching, which operates solely in the baseband at frequencies below ~50 MHz. It's impressive to see that the team has significantly reduced the power consumption compared to their earlier, seemingly identical version of the controller presented in [7].

While this work is highly impressive and certainly deserving of publication in a high-impact journal, I am not convinced that the novelty and breakthrough are significant enough to profoundly influence the field or facilitate spin qubit scaling beyond the current state-of-the-art, which would warrant publication in a prestigious journal such as Nature.

To be specific:

i) The manuscript demonstrates cryo-CMOS control of only two spin qubits, and even at this level, qubit coherence times are affected. There is no detailed discussion on how their approach could be scaled to control thousands of qubits while still operating at milli-kelvin temperatures without significantly impacting qubit coherence times or gate fidelities.

ii) While coherence times are not significantly affected by the cryogenic controller, qubit gate fidelities are not at state-of-the-art levels, and therefore not as sensitive to small control imperfections. Furthermore, the two-qubit gate fidelity is not reported. For example, in their previous work, much higher values were achieved, with single-qubit gate fidelities of 99.85% and two-qubit gate fidelities of 98.92% [19].

iii) Spin qubit readout is not addressed. The RFSET, operating at 400 MHz, exceeds the 50 MHz baseband generated by the cryo-CMOS controller. While a single SET or RFSET can be used to read out a few spin qubits, scaling to thousands or more qubits would still require a comparable number of signal lines to the number of qubits.

iv) Regarding the global MW field excitation, the authors claim that "fine time resolution is needed only to map out coherent oscillations," and that fixed-duration pulses for single gates like a $\pi/2$ gate are straightforward to implement on cryo-CMOS. However, I would argue that any controller capable of achieving high-fidelity single qubit gates (>99.9%) requires fine time resolution to precisely target carefully calibrated pulse durations. Additionally, this duration may vary between different qubits and would need to be recalibrated over time due to system drift. Therefore, significant time resolution is essential to ensure precise pulse duration timing in different situations.

v) While similar work has been published before [a,b], including by the authors themselves [7], it is unclear what specific advancements led to the breakthrough in power consumption beyond optimizing the controller and connecting it to the quantum chip. A more detailed description of the innovations that resulted in the ultra-low power consumption would be greatly appreciated, especially in comparison to their previous work where power consumption was significantly higher. For instance, this work uses FDSOI technology—did the authors utilize the back gate to lower the threshold voltage, and if so, by how much and on which components of the controller? Additionally, what was the operating voltage (V_{dd}) used?

a) Subramanian et al., VLSI 2024 <https://ieeexplore.ieee.org/document/10631530>

b) Bohuslavskiy et al., Commun Phys 7, 323 (2024).
<https://www.nature.com/articles/s42005-024-01806-3>

vi) In the manuscript, the cryo-CMOS chip is placed approximately 3 mm away from the quantum chip. Assuming the use of superconducting aluminum bond wires that are 50 μm thick, this would provide good thermal isolation between the hot CMOS and the cold quantum chip due to the high bond aspect ratio. However, as the system scales to thousands or more spin qubits, the number of control wire bonds would increase at least thousandfold. Would this architecture, with long wire bonds, still be practical? The authors also mention the prospect of lithographically defined chip-to-chip interconnects. If these interconnects are based on superconducting indium bumps, wouldn't the much smaller aspect ratio of the bumps compared to current wire bonds present a significant thermalization challenge?

Further comments:

vii) A key enabler for all-baseband cryo-CMOS control is the global microwave field combined with baseband pulses to shift the qubits' resonance closer to the global MW field for qubit excitation. The authors claim that such control is independent of simultaneously controlling other qubits and is scalable. However, it is unclear if this will hold true when higher single-qubit gate fidelities are required. For instance, when using a common global field for ten or more qubits, many more than two qubits may need to be excited simultaneously to maintain high qubit activity and avoid decoherence of idle qubits. Any small crosstalk between qubits during simultaneous baseband control could become problematic, especially as even higher single- and two-qubit gate fidelities are targeted for large-scale error correction. Can the authors justify that crosstalk between simultaneously excited qubits being closely resonant remains negligible even for high gate fidelities (e.g. for >99.9%)?

(Remarks to the Author)

The manuscript by Bartree et al. presents operations of silicon spin qubits by a cryo-CMOS control chip placed beside the qubit chip mounted on a milli-kelvin stage of a dilution refrigerator. The control chip holds gate biases on the qubits chip by DRAM-like charge-lock refresh cycle, and also generates baseband pulses according to digital signals sent from room temperature. The baseband pulses are utilized to switch qubit resonance frequencies and the exchange interaction. The authors compare qubit gate operations implemented by using the cryo-CMOS controller and room temperature electronics, and conclude the degradation in qubit performance from the cryo-CMOS is limited.

Cryo-CMOS controllers are highly demanded for scaling up quantum processors. The charge-locking circuit and baseband pulse generators are particularly important to reduce the number of signal lines from room temperature to low-temperature stages. While these functionalities have been reported by Pauka et al., *Nat. Electron.* 4, 64 (2021) (ref. 7) from the same group, this is the first application of them to actual gate operations of silicon spin qubits to my knowledge. A preceding report of cryo-CMOS control (ref. 20) is about different functionalities at higher temperature stage (4 K). This does not lower originality and significance of the conclusions of this manuscript, that is, a cryo-CMOS controller with limited impact on qubit operations at milli-kelvin and application of charge-locking and baseband pulse generation to silicon spin qubit operations.

However, I have some concerns about the authors' claims as follows:

1. Temperature of the qubit chip may be much higher than the mixing chamber temperature. As mentioned by the authors in Extended Data Fig. 3, electron temperature, at least, clearly deviates from the mixing chamber temperature. The temperature insensitivity of coherence times in the same figure also indicates that the qubit chip is as hot as 600 mK. If this temperature is more appropriate as the effective temperature of the qubit chip, the cryo-CMOS reported here should be referred to as a sub-kelvin cryo-CMOS rather than milli-kelvin cryo-CMOS. I note that, while the mixing chamber temperature is kept to tens mK, it is not important as its aim is to keep a qubit chip milli-kelvin.

2. Influence of electrical noise from the cryo-CMOS on qubit operations may be overlooked merely because of the noise power spectral density enhanced by the high effective temperature as reported in ref. 19. This point is mentioned by the authors in the discussion section (paragraph 'Our cryo-CMOS control chip comprises ...'). Then 'degradation in qubit performance from milli-kelvin CMOS is very limited' in conclusion looks an overstatement. Results of experiments at a surely low effective temperature or, instead, high two-qubit gate fidelity (or at least a pathway to obtain fault-tolerant fidelity with the cryo-CMOS) should be included in the manuscript for saying the degradation is very limited. Or, if the electron temperature during the baseline experiment only using room-temperature electronics (not included in the manuscript) is sufficiently low, the authors' conclusion can be justified.

3. The roles of the cryo-CMOS in the qubit operations are limited. While the baseband pulse generated by the cryo-CMOS is applied to the J gate, P1 and P2 gates are still controlled by room temperature electronics. If these plunger gates must be biased by room temperature electronics, the reduction of cables by using cryo-CMOS is moderate. The authors should elaborate the reason that the plunger gates need to be controlled by room temperature electronics, and clarify whether cryo-CMOS control of them is possible without increasing functionalities of the cryo-CMOS or not.

Overall, although the concept of the manuscript is impressive and should be considered in a high impact journal, the experimental results or explanations are insufficient to justify the conclusions of the manuscript. Thus, the current form of the manuscript does not convince me that the Nature is the best journal for this work.

I also have specific or minor comments that the authors should address:

4. Figure 2a shows data as a function of MW frequency and pulse time rather than MW power, while the main text says 'The shot-average readout signal as a function of MW power and frequency is shown in Fig. 2(a)' in the paragraph 'Turning to evaluate single qubit gates, ...'.

5. Figures 2d and 3b,d and Extended Data Figs. 1 and 5 are not mentioned. If these figures are not necessary, they should be removed.

6. Which qubit is controlled in Fig. 2a,b?

7. Can the authors specify the J-gate-bias sensitivity of exchange coupling at the points relevant to two qubit operations? It is an important parameter to discuss coupling between qubits and electrical noise.

8. Some datasets irrelevant to free-induction decay are referred as to 'FID data'.

9. The description 'The spins are initialized in the (T-) ground state via off-resonance relaxation' in the paragraph 'The global-MW scheme requires...' is not clear for me. Would you please elaborate more? Also, is this relaxation scheme identical with that used for the DCZ experiments?

10. Please specify MW frequencies for single qubit gates in the Methods section.

11. It is better to denote which qubit corresponds to each resonance feature in Extended Data Fig. 5a.

12. Neither number of measurement shots to calculate probabilities nor definition of error bars are not specified.

13. Misspelling in the Measurement Setup section: 'digital'.

14. For ref. 31, Laucht et al., *Sci. Adv.* 1, e1500022 (2015) is appropriate rather than *Nature Nanotechnol.* from the same group.

Reviewer comments:

Referee #1

(Remarks to the Author)

The authors have done a good job responding to the reviewers. In addition to the technical achievement presented in it, this work prompts several key questions for scaling, including how to best manage heat from cryo controllers, and how qubit variability affects cryo control. All in all, this is an important paper.

The one thing I am still not sure about involves this claim in the rebuttal "Here, we first re-establish this dependence by intentionally heating the refrigerator and measuring qubit performance. ... The measured qubit fidelity then, as a function of powering up each block, is well explained by just the increase in temperature." Is the data for the first part of this claim Extended Data Fig. 3? If I look at this data, it doesn't look like the qubit metrics change until the MC temperature exceeds 500 mK. On the other hand, the data of Fig. 2 show that the qubit metrics are changing when the MC temp goes up to just 25 mK. This is the main obstacle I have with the argument about temperature increases explaining the reduction in metrics. I must be missing something here, and I hope the authors can clear this up for me.

Referee #2

(Remarks to the Author)

The authors have successfully addressed most of the raised points. However, there are still some concerns that need to be highlighted:

- * The argument regarding power consumption scalability is difficult to accept, especially considering that a high activity factor will be necessary for a large-scale quantum computer. Nevertheless, I agree with the authors that there are still many possibilities to explore in terms of lowering the cryoCMOS temperature, particularly with on- and off-chip heat management.
- * The scalability of the given readout solution is unclear, given the large physical size of resonant elements. However, the community is actively exploring alternative solutions.
- * My main concern pertains to the global MW field control. The presented control system relies on this approach, which allowed the authors to design the controller to operate at baseband frequencies, thereby reducing power consumption. If it turns out that this approach is not scalable due to crosstalk in large quantum processors, the authors' cryo-control system will lose its merit. However, it is too early to determine this with certainty.

For these reasons, and considering that the demonstration was performed on a small scale, I remain unconvinced that the work is suitable for the prestigious journal Nature. Nonetheless, I am not strongly opposed to a positive outcome if the editors decide in favor of it.

Referee #3

(Remarks to the Author)

The authors have responded to all of my comments. Almost all of my comments are responded satisfactorily. Regarding the title mentioned in the response to "1. Temperature of the qubit chip may be ...", I prefer the authors' new suggestion "Spin Qubits with Scalable microwatt CMOS Control" to the original title. I still concern the authors' response to my comment "2. Influence of electrical noise from the cryo-CMOS ..." as follows:

1. Regarding "Lastly, we also remark ...", please elaborate the comparison in the manuscript with citing publications. If this comparison is appropriate, the smallness of noises contributed by the CMOS in the authors' claim is plausible.
 2. While the authors mention "electron temperatures of order 200 mK" citing Ref. 7, I could not find descriptions about electron temperatures with powering up the CMOS device in Ref. 7. Unless this is my overlooking, this electron temperature value is unpublished and the authors should not use this to reinforce their argument without showing the supporting data.
- Overall, I will be able to agree with the publication in Nature after the authors respond to my concern.

Response to Referee #1

1. *This paper presents a demonstration of spin-qubit control by a millikelvin CMOS chip. It is astonishing that this control method works at millikelvin temperatures. Most previous work in this direction has used control chips at elevated temperatures, but the charge-locking mechanisms allows the chip to work at millikelvin temperatures. This is an important step for large-scale quantum computing. My overall comment is that this paper would benefit from more precise arguments in support of the main claims. In my understanding, there are three main claims. First, the cryo-CMOS chip does not significantly reduce the qubit performance. Second, the cryo-CMOS chip does not introduce additional electrical noise (either through a fluctuating voltage or crosstalk), and third, the (slight) reduction in fidelity is likely due to excess heat and two-level-only pulse control.*

Response: We thank the reviewer for their thoughtful comments and careful reading of our manuscript. In addressing their comments, we have modified the paper to include new, specific details and enhance the overall clarity.

2. *About the first claim, the RB results support the claim that the single-qubit gate fidelities are not reduced. However, the measurements of state preparation, readout, and two-qubit gate performance are based on qualitative metrics, like visibility, and the decay of two-qubit oscillations. Why were SPAM and 2Q fidelities not measured? If these measurements are not easy to perform, and if the authors agree that these would be helpful, it would be good to state this in the conclusion.*

Response: This is a good question that warrants a detailed discussion and further adjustment of the paper, particularly with respect to how to improve future experiments. For this particular device, a complete quantitative assessment of state preparation and measurement (SPAM) and 2Q fidelities was indeed not possible using our prototype cryo-CMOS platform. As the Referee notes, we do still perform comparisons based on visibility and decay of 2Q coherent oscillations. Detailed protocols such as RBM and GST were also performed using the cryo-CMOS platform, however for this particular device the two-level control resulted in data that does not reflect the true device fidelities. Unfortunately, as is sometimes the case, this device had little overlap between the voltage levels for the exchange (J) gate and those needed for SPAM characterization (owing to the different tunnel rates used in each process). On the other hand, some devices do exhibit strong overlap between the SPAM voltage parameters and the control voltage window (for both 1 and 2 qubit operations), and for such devices the current cryo-CMOS architecture with two-level control would be sufficient to perform more sophisticated measurements, including 2-qubit RBM and GST to further assess fidelity metrics.

Looking to the future then, we are thus presented with an important trade space and decision: increase the sophistication and number of voltage levels in the control architecture (at the expense of power dissipation and heat), or improve qubit device variability and tunability? Likely both are needed in the end. On the qubit side, we do know that it is common to have strong overlap between parameters needed for SPAM and those for control. Improving this overlap ultimately helps with scalability of the architecture as it reduces the burden on the

control systems. In future work we will increase the number of available cryo-CMOS pulses, mindful that qubit device design can also be significantly improved to ensure the relevant tunnel rates are in an appropriate range. We have added a discussion to the paper stating clearly the reason quantitative SPAM metrics were not extracted for this qubit device using cryo-CMOS [mod1 – pg. 4 para. 4].

3. About the second claim, certainly the fact that the qubit metrics do not diminish significantly compared to RT control is a strong argument. However, I wonder about the reduction in the metrics as more items are powered up. How do we know this is not due to electrical noise? Looking at the data of Extended Data Fig. 4f, for example, what is the mechanism behind the increase in the noise as the oscillator is turned on?

Response: Previously published investigations (eg, Ref. [19]) have examined the dependence of qubit coherence and fidelity on temperature. Here, we first re-establish this dependence by intentionally heating the refrigerator and measuring qubit performance. Performing the same measurement again but now using the CMOS chip as an effective heater, we then look to see if qubit performance is degraded beyond what can be accounted for from heat alone. In essence, we independently know the relationship between qubit fidelity and temperature and the temperature change that occurs when each CMOS block is powered up. The measured qubit fidelity then, as a function of powering up each block, is well explained by just the increase in temperature. If there were additional electrical noise on top of this temperature dependence it would thus lead to a further reduction in qubit fidelity.

As to the mechanism underpinning the temperature dependence, previously work has identified thermal noise as the prime candidate. Heating the qubit chip increases the Johnson-Nyquist voltage fluctuations from all components having a real-part of the impedance. Our data in Extended Data Fig. 4f further support this mechanism. Here we use a qubit as a spectrometer of noise in a band selected by CPMG pulse sequences. Powering-up the CMOS oscillator, we observe a slight increase in kHz white noise, rather than narrow-band, frequency spurs that might be expected from clock-bleed and (sub-harmonic) crosstalk effects. So, again, the CMOS oscillator generates some heat, which increases the temperature of the qubit chip and in proportion, leads to an increase in the white thermal noise that degrades qubit fidelity. We believe that electrical noise (such as the digital ‘hash’ generated by the rising and falling edges of switching transistors) is unlikely to mimic Johnson-Nyquist noise in perfect proportion to the increase in temperature. In fact, even if such electrical noise somehow mimicked a thermal noise source, this electrical noise would appear on-top of the thermal noise background, ie, in addition to the thermal noise contribution we can already account for. That we do not detect such electrical noise (especially when performing sensitive 2-qubit J-gate operations) is a surprising and key result of our paper. We have adjusted the manuscript to emphasize this point [mod2 – pg. 5 para. 1].

4. About the third claim, what exactly is the argument in support of this? I can see that TMC increases as more items are powered up, and I can imagine that this could increase, for example, the charge noise level experienced by the qubit as the qubit-chip temperature increases. But I could also imagine,

for example, that the noise from the cryo-CMOS device also increases (see above). I get the impression that Extended Data Fig. 3 is supposed to help with this argument, but I can't quite make the link.

Response: We understand the Referee's interest in these details, as they are surprising and have important implications for the scalability of spin qubit architectures based on millikelvin CMOS control. As discussed above, as each CMOS block is powered up we firstly see a clear dependence in the qubit fidelity that corresponds exactly with what is observed by simply increasing the temperature (ie, by using a heater). Going beyond this, it is also possible to look for evidence of electrical noise and its impact on qubit performance in a regime where temperature is held near constant. For example, by reconfiguring the operation of the CMOS circuit but keeping the overall power density the same.

In Fig. 2c, for instance, we measure T_2^* with all 30 CLFG cells pulsing and compare it to the case when the cells are deactivated. Since measuring T_2^* is a wideband probe, sensitive to high frequency components, low frequency noise, drift, offset, etc, it provides a careful means of measuring additional electrical noise. Even when pulsing all 30 cells however, we see no difference in T_2^* . Importantly, since the power of each cell is only 20nW/MHz (for 100 mV pulses), the associated increase in qubit temperature is very small. Similar measurements are reported in the data shown in Fig. 4 and the extended data in Fig. 5 where we probe the two-qubit CZ and DCZ T_2 exchange oscillations as a function of different configurations of the CMOS circuits. Again, we do not observe degradation in the fidelity of qubit operations from additional electrical noise. Lastly, we underscore that comparing 1- and 2-qubit data taken using room temperature control to data taken using cryo-CMOS also supports the claim that any electrical noise generated by the CMOS platform is sufficiently low to be undetectable via such a comparison.

5. About the claim that two-level-only control increases SPAM errors, what is the argument in support of this? For example, do the authors have a model that shows if they include only two-level control, SPAM errors increase, and can they reproduce the observed data with the model?

Response: As discussed above, for this particular device it was not possible to find overlapping regions of voltage space that enable optimal state preparation, control, exchange, and readout. That is, the optimal tunnel rates required for these operations are rather different. Attempting to cover these regions of voltage space with just two-level control leads to an increase in SPAM error. Importantly, if room temperature control is limited to two-level pulsing, then correspondingly, the same increase in SPAM error is observed. Although this is not a detailed model, it does validate our argument and allow the data to be reproduced under the assumptions of two-level control.

Figure 4 provides evidence for this effect. In (a), an optimized, many level pulse sequence is generated with RT electronics, with different voltage levels used for state preparation, control, exchange, and readout. When switching to two-level control (b) and (c), either using RT electronics or cryo-CMOS, state preparation, readout, and exchange are grouped. Since

optimized qubit control requires a tunnel rate much too low for any other operation, the readout visibility is degraded, compared to (a). Although in this qubit device the tunnel rates for control and SPAM were significantly different, we emphasize that it is not uncommon to find devices that have comparable tunnel rates. For such devices the two-level control can accommodate the full spectrum of operations. Certainly, for future experiments we will increase the number of voltages levels available with the cryo-CMOS platform to alleviate this issue.

6. *Smaller comments include the following. What is the uncertainty on the value of $T2^*$ in Fig. 3d? Is this $T2^*$ value meaningfully lower than the $T2^*$ values presented in Fig. 2c? If so, why?*

Response: We thank the Referee for catching this and have added the uncertainties. The slight reduction in $T2^*$ is likely due to drift in Larmor frequency. These can be compensated for via routine calibration. [mod 3 – fig. 3]

7. *Are the RT/cryo-CMOS comparisons done in the same cooldown? If so, how? Why are the frequencies in Fig. 4c different from each other?*

Response: Thanks for this question – it raises an important point. Indeed, all measurements are done on the same cooldown. We were concerned that any thermal cycling of the device may negate our objective for an apples-to-apples comparison between room temperature control and CMOS, and thus avoided cycling the refrigerator that can lead to differences in the disorder potential of the quantum device.

Comparison between control approaches is made using the ‘passthrough mode’ of the CMOS chip that simply connects room temperature inputs to the output of the CMOS chip (see Fig. 1c). Like the comment above about an apples-to-apples comparison, this technique avoids different bonding and packaging arrangements which would be otherwise necessary to accommodate both room temperature and cryo-control. It further allows control of exactly the same gates, eg, the exchange gate for two-qubit operations using either room temperature or cryo-control. Such a comparison would be impossible without use of the passthrough setup. The slight differences in frequencies in Fig. 4c result from slight differences between the charge-lock voltage sourced from the cryo-CMOS verse room temperature. This can be accounted for in calibration. We have added details in the Methods section to clarify these aspects. [mod4 – pg. 6 para. 5]

8. *The electron temperature seems quite high at 850 mK. Do the authors expect this is because of the introduction of the cryo-CMOS chip?*

Response: Unfortunately, the present experimental set up suffered from poor thermalization of the cables from room temperature which likely contribute to the increased electron temperature. Ordinarily, we would warm up the fridge and address several of these technical issues, however, wanting to avoid thermal cycling to make an apples-to-apples comparison we elected to keep all aspects of the wiring setup constant for the whole experiment.

While electron temperature was elevated, importantly, the temperature of the mixing chamber stage of the refrigerator remained below 25 mK throughout the experiment. Increases in electron temperature well above the refrigerator then are the result of poor thermal contact between the electrons and (ultimately) the refrigerator. We believe there is substantial opportunity to improve this contact, and to further isolate heat from the CMOS. In our previous cryo-CMOS experiments with similar power densities, detailed in Ref. [Pauka, Nature Electronics], electron temperatures around 200 mK were possible using a much more sophisticated thermal management approach. Further unpublished data (taken in a very different context) suggests that electron temperature of order 100 mK is possible using superconducting interconnects for enhanced thermal isolation. Finally, we note that one of the advantages of SiMOS spin qubits is that they maintain fidelities even up to 1 kelvin (Ref. 17), alleviating some of the burden of engineering lowest electron temperatures.

9. Are there prospects for more-than-two-level control?

Response: Most certainly yes! Our preliminary designs and circuit models of future chips suggest it is possible to reduce the power density by as much as 100 X, furthering the scalability of the control approach. This reduction in power density also opens the prospect of not only increasing the resolution of the pulses but adding significantly more functionality in many aspects of the cryogenic system on chip (SoC). For instance, separate optimized circuits can be used for detailed benchmarking and calibration, switching over to optimized circuits for quantum ‘runtime’ execution. With the publication of the present work, we believe the spin qubit community (and other qubit platform communities also) will now direct significant attention to heterogeneously integrated cryo-CMOS control as a means of enabling scale-up of these systems. We expect many new advances to be reported now this approach has been validated with real qubits.

Response to Referee #2

1. The manuscript demonstrates cryo-CMOS control of only two spin qubits, and even at this level, qubit coherence times are affected. There is no detailed discussion on how their approach could be scaled to control thousands of qubits while still operating at milli-kelvin temperatures without significantly impacting qubit coherence times or gate fidelities.

Response: Our paper presents data demonstrating how baseband control via a cryo-CMOS platform can be combined with a microwave global field to enable scalable control to 1000s of qubits and beyond. The small impact on qubit fidelity observed in the current experiment stems from heat associated with the baseline CMOS circuit blocks (power-up, clock, finite-state machine, etc). Importantly, these baseline blocks are not duplicated with every qubit or qubit gate electrode but are rather a one-off hit as the circuits are powered up to control many qubits. In contrast, adding more qubits requires adding more CLFG cells in proportion, with each cell dissipating ~ 20 nW/MHz to generate pulses with 100 mV amplitudes. Thus, in principle, 1000 CLFG cells would contribute 20 micro-Watts of power per MHz, under the condition that every gate was simultaneously pulsing. The charge-lock mechanism we

demonstrate, which is used to generate dc-biases dissipates negligible power. Thus, considering the cooling power of a commercial refrigerator at ~ 0.3 K or so can exceed several milli-Watts, combined with the potential for high fidelity operation at elevated temperatures, control of many thousands of qubits appears possible in terms of the power constraints.

In the paper we test the scalability of this approach by characterizing qubit performance as each of the 30 CLFG cells are powered up. Even with all 30 cells pulsing, we do not see heating or degradation in qubit performance, below the baseline circuit blocks such as the clock and FSM (as shown in Figs. 2c, 4d, and Extended Data Figs. 4c and 5i). Further, as noted in the paper, we believe there is considerable opportunity to lower the overall power dissipation of the CMOS architecture in future designs, and to further improve the thermal management to limit the heat flow from CMOS to qubits. Given the data in hand and these improvements, we see no obvious barrier now to scaling this control approach to 1000s of qubits and beyond.

2. While coherence times are not significantly affected by the cryogenic controller, qubit gate fidelities are not at state-of-the-art levels, and therefore not as sensitive to small control imperfections. Furthermore, the two-qubit gate fidelity is not reported. For example, in their previous work, much higher values were achieved, with single-qubit gate fidelities of 99.85% and two-qubit gate fidelities of 98.92% [19].

Response: The focus of the present work is the scalable control approach, making use of qubits with currently typical, rather than record fidelities. Importantly, the fidelity of the current device does not achieve state-of-the-art levels when controlled using standard room temperature control electronics but remains at the typical level regardless of the control approach.

We agree that as qubit fidelities are improved, new noise sources may be revealed below the current noise floor and must be investigated and addressed. This is an ongoing iterative process. As qubit fidelity is further improved, we expect that the noise from the CMOS control will at some point be revealed as a limiting factor and can then be addressed, similar for other sources of noise, such as magnetic field drift, TLS, stray radiation, etc. For now, the noise from the cryo-CMOS control is sufficiently low to not degrade the performance of qubits below today's typical fidelities. We have added a comment in the manuscript to this point. [mod5 – pg. 4 para. 4]

To speculate about the future for a moment, we note that MOS spin qubit fidelities appear to be presently limited by charge noise and thermal noise associated with all device structures that constitute a real-part of the impedance of the environment. It is possible that a significant fraction of this thermal noise can be suppressed by cooling the control circuits and all related interconnect and packaging structures. We see some hints already in our data that cryo-CMOS may thus outperform room temperature control approaches in terms of thermal noise and consequently qubit fidelity. As such, the improvements in qubit fidelities beyond current state-of-the-art levels may come from developing cryogenic control approaches, of which our present prototype is a first step. We realize it is too early to make such a claim in the

present manuscript but thought it helpful to make this suggestion here in discussion with the Referee.

3. Spin qubit readout is not addressed. The RFSET, operating at 400 MHz, exceeds the 50 MHz baseband generated by the cryo-CMOS controller. While a single SET or RFSET can be used to read out a few spin qubits, scaling to thousands or more qubits would still require a comparable number of signal lines to the number of qubits.

Response: We thank the Referee for raising the question of scalable readout, which we believe may likely also occur to a general reader. We have thus now added a note in the conclusion of the manuscript. [mod6 – pg. 6 para. 2]

Our cryo-CMOS control architecture is not intended to address the challenges posed by performing readout at scale. Rather, on the control side, CMOS is well placed to generate large, volt-scale signals, whereas readout requires the detection of nano-volt level signals on fast timescales. Although, in principle, generating carrier frequencies ~ 400 MHz is straightforward with CMOS, we believe this hardly addresses the key issues associated with (near quantum-limited) amplification, and the challenges posed by having the number of signal wires comparable to the number of qubits (as noted by the Referee). Alternatively, our approach is to pair cryo-CMOS control with a scalable readout architecture based on frequency multiplexed reflectometry (See Ref. 38). In such an approach, 1000 or so readout channels can potentially be accommodated on a single readout lane comprising amplifier, isolator, and transmission line, (assuming a single channel bandwidth of 10 MHz and a lane bandwidth of several GHz). Here, room temperature direct digital synthesis is used to produce a comb of frequencies from a single DAC and is acquired by a single, wideband ADC). We believe this approach addresses the challenges of scalable readout and is highly compatible with our scalable approach based on cryo-CMOS. A full demonstration of a complete scalable control and readout approach will be the subject of future work.

4. Regarding the global MW field excitation, the authors claim that "fine time resolution is needed only to map out coherent oscillations," and that fixed-duration pulses for single gates like a $\pi/2$ gate are straightforward to implement on cryo-CMOS. However, I would argue that any controller capable of achieving high-fidelity single qubit gates ($>99.9\%$) requires fine time resolution to precisely target carefully calibrated pulse durations. Additionally, this duration may vary between different qubits and would need to be recalibrated over time due to system drift. Therefore, significant time resolution is essential to ensure precise pulse duration timing in different situations.

Response: We agree with the Referee that precision in the control is needed, but note firstly that pulse amplitude, offset, and pulse width in time can all be used to tune the effective angle of rotation of the qubit state vector. We also draw the distinction between precision and resolution, ie. dynamic range. For instance, with precise calibration of the two voltage levels, it is possible to implement high-fidelity control as long as these voltage levels are reproducible. Certainly, our CMOS approach also allows extremely fine control of the clock frequency, either programmed using the on-board oscillator, or externally triggered. Moreover,

the data suggest that tuning and calibration of qubit operations can be achieved by precise control of the dc-offset voltages, set by the CMOS charge locking circuits rather than requiring high resolution pulses. We have added a comment about how our control architecture shifts the requirements for high precision calibration and tuning onto the dc-voltage levels rather than requiring high resolution pulses [mod7 – pg. 4 para. 2].

5. *While similar work has been published before [a,b], including by the authors themselves [7], it is unclear what specific advancements led to the breakthrough in power consumption beyond optimizing the controller and connecting it to the quantum chip. A more detailed description of the innovations that resulted in the ultra-low power consumption would be greatly appreciated, especially in comparison to their previous work where power consumption was significantly higher. For instance, this work uses FDSOI technology—did the authors utilize the back gate to lower the threshold voltage, and if so, by how much and on which components of the controller? Additionally, what was the operating voltage (V_{dd}) used?*

a) Subramanian et al., VLSI 2024 <https://ieeexplore.ieee.org/document/10631530>

b) Bohuslavskyi et al., Commun Phys 7, 323 (2024).

<https://www.nature.com/articles/s42005-024-01806-3>

Response: Our approach to minimizing the power density of the control platform firstly leverages many of the well-known techniques in low-power (room temperature) CMOS design. We have modelled and examined many different circuit implementations to arrive at the lowest power designs tailored for operation at cryogenic temperatures. Beyond these techniques, we strongly exploit the highly suppressed leakage currents of transistors at these temperatures and reduction in some parasitics. As the Referee suggests, a further key enabler in low power operation is the use of a back gate (or body bias) in FDSOI technology. In fact, our circuits allow for different back gate bias for different circuit blocks on the same chip (see Ref. below), which allows for significant configurability to optimize power density and performance. In general, we lower V_{dd} to its lowest value needed for circuit functionality. This can be different for different circuit blocks. In comparison to previous work, we note that the specific power consumption depends of course on exactly what circuit blocks are powered up, clock rate, memory usage, and utility factor. Previous work explored situations where all circuit blocks are simultaneously operated and pushed to maximum performance. Here, not all blocks are needed to control qubits. In terms of the specifics, all circuit blocks were measured to be functional within the range DD1P0 = 0.65V, N/PMOS Backgate=1.5/-1.5V, VDD1p8 = 1.2V-1.6V. Under these adjustments power dissipation is then 5.2x lower than the nominal setting. It may also be of interest to note that the CLFG circuit ensures that 1/f noise from the dc charge-locking transistor is blocked, and 1/f noise from the fast-gating transistors are high-pass filtered – this way, 1/f noise does not directly translate to the output. We have added more of these details to the manuscript. [mod8 – pg. 6 para. 7]

Reference: Y. Yuan, K. Das. A. Moini, D. J. Reilly, IEEE Solid-State Circuits Letters 3, 186 (2020).

6. In the manuscript, the cryo-CMOS chip is placed approximately 3 mm away from the quantum chip. Assuming the use of superconducting aluminum bond wires that are 50 μm thick, this would provide good thermal isolation between the hot CMOS and the cold quantum chip due to the high bond aspect ratio. However, as the system scales to thousands or more spin qubits, the number of control wire bonds would increase at least thousandfold. Would this architecture, with long wire bonds, still be practical? The authors also mention the prospect of lithographically defined chip-to-chip interconnects. If these interconnects are based on superconducting indium bumps, wouldn't the much smaller aspect ratio of the bumps compared to current wire bonds present a significant thermalization challenge?

Response: We thank the Referee for this important discussion about thermal management, which we believe is a critical new area of development for the field. Indeed, we envisage a packaging arrangement that leverages fine-pitch superconducting indium bump bonds, as the Referee suggests. This issue of the small aspect ratio of the bump can be addressed by incorporating a longer length of (lithographically defined) superconducting track on a multi-chip module (MCM), prior to the indium bump to provide thermal isolation. In truth however, the requirement for the qubits to operate in magnetic fields of order 0.5 Tesla will likely drive the need for alternate superconductors beyond aluminium (in the present experiment the critical field of the bond-wires is already exceeded). Niobium materials (eg NbTiN) may be suitable, although there is appreciable thermal conductivity from sub-gap states. Materials such as lead appear to be ideally placed for creating thermally isolating interconnects and future publications will report details on this topic.

7. A key enabler for all-baseband cryo-CMOS control is the global microwave field combined with baseband pulses to shift the qubits' resonance closer to the global MW field for qubit excitation. The authors claim that such control is independent of simultaneously controlling other qubits and is scalable. However, it is unclear if this will hold true when higher single-qubit gate fidelities are required. For instance, when using a common global field for ten or more qubits, many more than two qubits may need to be excited simultaneously to maintain high qubit activity and avoid decoherence of idle qubits. Any small crosstalk between qubits during simultaneous baseband control could become problematic, especially as even higher single- and two-qubit gate fidelities are targeted for large-scale error correction. Can the authors justify that crosstalk between simultaneously excited qubits being closely resonant remains negligible even for high gate fidelities (e.g. for >99.9%)?

Response: We agree with the Referee that base-band control schemes are sensitive to crosstalk. Moreover, given the capacitive nature of these qubit devices, crosstalk is an unavoidable aspect of the architecture that must be fundamentally addressed at the foundation of the control scheme. Presently the approach to address this makes use of so-called virtual gates, where the various cross capacitances of the gate electrodes are calibrated and incorporated using software such that compensating voltages are always applied to nearby gates to null capacitive coupling. It appears that such techniques are also well suited for multi-qubit operation with low overhead. A next step involves qubit device design to include the use of ground planes, fencing vias, and guard structures to isolate electric fields. We also note that global control architectures generally make use of dressed-qubit approaches where all idle qubits are quasi-degenerate and kept on-resonance,

undergoing continuous dynamical decoupling sequences to null low-frequency noise and offsets (see references below). Finally, we also point out that since at the physical qubit level the system will be executing an error correcting code, eg, a Surface Code, the control signals play out in a constantly repeating and deterministic pattern, with half the qubits doing the same operation each cycle. With prior knowledge of these signals, qubit layout, pulse compensation, and system optimization can be structured to minimize crosstalk in comparison to circuits where there are many more non-deterministic complex operations happening simultaneously.

Lastly it is important to say, that this is a general issue for spin qubits irrespective of the control approach. The challenges exist equally for conventional room temperature approaches as well as scalable cryo-CMOS architectures.

See Refs:

Hansen et al., Nat. Comm. 7656 (2024):

Entangling gates on degenerate spin qubits dressed by a global field | Nature Communications

Cifuentes et a., Nat. Comm. 4299 (2024):

Bounds to electron spin qubit variability for scalable CMOS architectures | Nature Communications

Response to Referee #3:

The manuscript by Bartee et al. presents operations of silicon spin qubits by a cryo-CMOS control chip placed beside the qubit chip mounted on a milli-kelvin stage of a dilution refrigerator. The control chip holds gate biases on the qubits chip by DRAM-like charge-lock refresh cycle, and also generates baseband pulses according to digital signals sent from room temperature. The baseband pulses are utilized to switch qubit resonance frequencies and the exchange interaction. The authors compare qubit gate operations implemented by using the cryo-CMOS controller and room temperature electronics, and conclude the degradation in qubit performance from the cryo-CMOS is limited.

Cryo-CMOS controllers are highly demanded for scaling up quantum processors. The charge-locking circuit and baseband pulse generators are particularly important to reduce the number of signal lines from room temperature to low-temperature stages. While these functionalities have been reported by Pauka et al., Nat. Electron. 4, 64 (2021) (ref. 7) from the same group, this is the first application of them to actual gate operations of silicon spin qubits to my knowledge. A preceding report of cryo-CMOS control (ref. 20) is about different functionalities at higher temperature stage (4 K). This does not lower originality and significance of the conclusions of this manuscript, that is, a cryo-CMOS controller with limited impact on qubit operations at milli-kelvin and application of charge-locking and baseband pulse generation to silicon spin qubit operations.

Response: We appreciate the thoughtful comments and careful reading of our work. Below, we address specific comments.

However, I have some concerns about the authors' claims as follows:

1. *Temperature of the qubit chip may be much higher than the mixing chamber temperature. As mentioned by the authors in Extended Data Fig. 3, electron temperature, at least, clearly deviates from the mixing chamber temperature. The temperature insensitivity of coherence times in the same figure also indicates that the qubit chip is as hot as 600 mK. If this temperature is more appropriate as the effective temperature of the qubit chip, the cryo-CMOS reported here should be referred to as a sub-kelvin cryo-CMOS rather than milli-kelvin cryo-CMOS. I note that, while the mixing chamber temperature is kept to tens mK, it is not important as its aim is to keep a qubit chip milli-kelvin.*

Response: Ultimately, it is the power density of the CMOS relative to the cooling power of refrigerator that sets the limit on temperature of the qubit chip. Here, the total power of the control system remains in the 10s of microWatts, maintaining the refrigerator at 25 mK or so. Electron temperature of the qubits, can of course, exceed this baseline set by the fridge, owing to the limited thermal coupling between the electrons and phonons of the lattice at low temperatures (dependent on qubit device details). Although in our current experiment this thermal coupling is poor, previous work [Ref. 7] achieved electron temperatures close to 200 mK, and we believe this can be significantly improved in future. Ultimately, in a scaled-up system it is possible to imagine separate cooling refrigeration systems for qubits and control.

Moreover, we note that with recent progress in 'hot' silicon spin qubits [Ref. 19], the target temperature of a scaled-up system is of order 1 kelvin, highly compatible with the power density of our CMOS control architecture. In such a scaled-up system the relevance of electron temperature is an open question, since the thermal distribution of electrons in the 2D reservoirs (needed for loading and measuring qubits) may not directly affect qubit performance.

In terms of the title suggested by the Referee, we are happy to make this modification, however, given previous work has shown that electron temperatures ~ 200 mK are possible using this approach, we leave this decision to the Editor's discretion. Another possibility is "Spin Qubits with Scalable microWatt CMOS Control".

2. *Influence of electrical noise from the cryo-CMOS on qubit operations may be overlooked merely because of the noise power spectral density enhanced by the high effective temperature as reported in ref. 19. This point is mentioned by the authors in the discussion section (paragraph 'Our cryo-CMOS control chip comprises ...'). Then 'degradation in qubit performance from milli-kelvin CMOS is very limited' in conclusion looks an overstatement. Results of experiments at a surely low effective temperature or, instead, high two-qubit gate fidelity (or at least a pathway to obtain fault-tolerant fidelity with the cryo-CMOS) should be included in the manuscript for saying the degradation is very limited. Or, if the electron temperature during the baseline experiment only using room-temperature electronics (not included in the manuscript) is sufficiently low, the authors' conclusion can be justified.*

Response: We agree that it is always possible to further hunt for new noise contributions after the leading source is suppressed and this will be an on-going aspect of refining the system as it scales. For now, the least we can say is that any noise contributed by the CMOS is already below the effective degradation produced by a ~ 600 mK electron temperature, ie, below 50

ueV. We believe such a statement is already significant, and as discussed above, since previous work has achieved electron temperatures of order 200 mK, we believe it is unlikely that an elevated electron temperature is masking some small additional noise contribution. Rather, in contrast to a white noise source, it is hard to imagine how an elevated electron temperature would mask qubit degradation due to, for instance, crosstalk frequency spurs from, say CMOS clock bleed. That such frequency components are, at worst, below the thermal 600 mK background is a surprising and important result of this work. It may also be of interest to note here that the CLFG circuit ensures that $1/f$ noise from the dc charge-locking transistor is blocked, and $1/f$ noise from the fast-gating transistors are high-pass filtered – this way, $1/f$ noise does not directly translate to the output.

Lastly, we also remark that the qubit fidelities and two-qubit coherence times measured with CMOS compare well to typical values measured in many previous experiments that exhibit low electron temperatures. The two-qubit coherence time with CMOS is comparable to devices that exhibit fault-tolerant fidelity.

3. The roles of the cryo-CMOS in the qubit operations are limited. While the baseband pulse generated by the cryo-CMOS is applied to the J gate, P1 and P2 gates are still controlled by room temperature electronics. If these plunger gates must be biased by room temperature electronics, the reduction of cables by using cryo-CMOS is moderate. The authors should elaborate the reason that the plunger gates need to be controlled by room temperature electronics, and clarify whether cryo-CMOS control of them is possible without increasing functionalities of the cryo-CMOS or not.

Response: The charge-lock fast gate (CLFG) functionality is demonstrated here with both the J-gate and also the B-gate enabling both dc-bias and baseband pulses. There is no requirement for any room temperature control and it is in principle possible for all gates to be controlled direct from the CMOS platform. Here, we have made choices to bond some gates to room temperature electronics in order to facilitate comparison in performance of control approaches (without thermal cycling the device). These connections also allow potential noise on the sensitive J-gate to be isolated to the CMOS system, rather than a convolution of potential noise sources. We emphasize that it is the J-gate that is the most sensitive to electrical noise. We thank the Referee for raising this question and have added a comment in the manuscript to better explain these details. [mod9 – pg. 3 para. 2].

I also have specific or minor comments that the authors should address:

4. Figure 2a shows data as a function of MW frequency and pulse time rather than MW power, while the main text says ‘The shot-average readout signal as a function of MW power and frequency is shown in Fig. 2(a)’ in the paragraph ‘Turning to evaluate single qubit gates, ...’.

Response: We thank the Referee for the catch and have updated the manuscript to read: “...as a function of MW pulse time...” [mod10 – pg. 3 para. 2]

5. *Figures 2d and 3b, d and Extended Data Figs. 1 and 5 are not mentioned. If these figures are not necessary, they should be removed.*

Response: We thank the Referee for catching the absence of reference to the figures. We now note the relevance of these data sets in the Experimental Results and Demonstrations section, and also in the Measurement Setup section of Methods. [mod11 – pg. 3, 4, 5, 6, para. 5, 2, 1, 3 respectively]

6. *Which qubit is controlled in Fig. 2a, b?*

Response: We thank the Referee for this catch. We have added a note to identify qubit 1. [mod12 – fig. 2]

7. *Can the authors specify the J-gate-bias sensitivity of exchange coupling at the points relevant to two qubit operations? It is an important parameter to discuss coupling between qubits and electrical noise.*

Response: We agree and thank the Referee for calling this out. We have added a note that the J-gate drives two-qubit exchange at a sensitivity of 25 dec/V. [mod13 – extended data fig. 5]

8. *Some datasets irrelevant to free-induction decay are referred as to 'FID data'*

Response: Again, we thank the Referee for catching this typographically error and have corrected the manuscript. [mod14 – pg. 5 para. 1]

9. *The description 'The spins are initialized in the (T-) ground state via off-resonance relaxation' in the paragraph 'The global-MW scheme requires...' is not clear for me. Would you please elaborate more? Also, is this relaxation scheme identical with that used for the DCZ experiments?*

Response: We thank the Referee for identifying this confusing sentence and have revised it to improve clarity, including adding references. Indeed, this relaxation scheme is identical for DCZ operation. "Here, the spins are initialized to a $T^{\wedge-}$ ($|\downarrow\downarrow\rangle$) state off-resonance via a detuning pulse to a relaxation hot-spot." [mod15 – pg. 4 para. 2].

10. *Please specify MW frequencies for single qubit gates in the Methods section.*

Response: We thank the Referee for this catch and have added, "Single qubit gates are operated at a MW frequency of 13.9 GHz." [mod16 – pg. 6 para. 3]

11. *It is better to denote which qubit corresponds to each resonance feature in Extended Data Fig. 5a.*

Response: We agree and have modified the figure. [mod17 – extended data fig. 5a]

12. *Neither number of measurement shots to calculate probabilities nor definition of error bars are not specified.*

Response: We thank the Referee for identifying this oversight, which has now been corrected. Measurement shots and error bar definitions have been added. [mod18 – fig. 2-4, extended data figs. 3-5]

13. *Misspelling in the Measurement Setup section: 'digital'.*

Response: We thank the Referee and have fixed this typo. [mod19 – pg. 6 para. 3].

14. *For ref. 31, Laucht et al., Sci. Adv. 1, e1500022 (2015) is appropriate rather than Nature Nanotechnol. from the same group.*

Response: We thank the Referee and have added the Reference. [mod 20 – pg. 4 para. 1]

Bartee et al., Response to Referees: Round 2

Referee 1

The authors have done a good job responding to the reviewers. In addition to the technical achievement presented in it, this work prompts several key questions for scaling, including how to best manage heat from cryo controllers, and how qubit variability affects cryo control. All in all, this is an important paper.

The one thing I am still not sure about involves this claim in the rebuttal “Here, we first re-establish this dependence by intentionally heating the refrigerator and measuring qubit performance. ... The measured qubit fidelity then, as a function of powering up each block, is well explained by just the increase in temperature.” Is the data for the first part of this claim Extended Data Fig. 3? If I look at this data, it doesn’t look like the qubit metrics change until the MC temperature exceeds 500 mK. On the other hand, the data of Fig. 2 show that the qubit metrics are changing when the MC temp goes up to just 25 mK. This is the main obstacle I have with the argument about temperature increases explaining the reduction in metrics. I must be missing something here, and I hope the authors can clear this up for me.

Response to Referee 1: We thank the Referee for their thoughtful and positive comments on the paper. The temperature dependence of the qubit metrics can be explained by the difference in the electron temperature of the qubit device and the mixing chamber thermometer anchored to the refrigerator. For a given power dissipation of a few 10s of microwatts, the impact on the refrigerator temperature is very small, but local heating of the electrons can occur. Certainly, this can be mitigated in future by enhanced approaches to thermal management, bringing the cooling power at the CMOS chip closer to the cooling power of the mixing chamber stage of the refrigerator. As the Referee notes, the qubit metrics don’t change much until the mixing chamber is heated above 500 mK since this is approximately the electron temperature in the device. We hope this explanation addresses the question raised. We have adjusted the manuscript to make this point clearer with the statement, in the Discussion section, 2nd para:

“This arrangement can lead to elevated electron temperatures in the quantum device (see Extended Data Fig. 3) even when the refrigerator remains cold (see Fig. 2d) and is the likely explanation for the small impact we observe in qubit fidelity when the largest CMOS circuits are powered up at the highest clock rates.”

Referee 2

The authors have successfully addressed most of the raised points. However, there are still some concerns that need to be highlighted:

** The argument regarding power consumption scalability is difficult to accept, especially considering that a high activity factor will be necessary for a large-scale quantum computer. Nevertheless, I agree with the authors that there are still many possibilities to explore in terms of lowering the cryoCMOS temperature, particularly with on- and off-chip heat management.*

** The scalability of the given readout solution is unclear, given the large physical size of resonant elements. However, the community is actively exploring alternative solutions.*

** My main concern pertains to the global MW field control. The presented control system relies on this approach, which allowed the authors to design the controller to operate at baseband frequencies, thereby reducing power consumption. If it turns out that this approach is not scalable due to crosstalk in large quantum processors, the authors' cryo-control system will lose its merit. However, it is too early to determine this with certainty. For these reasons, and considering that the demonstration was performed on a small scale, I remain unconvinced that the work is suitable for the prestigious journal Nature. Nonetheless, I am not strongly opposed to a positive outcome if the editors decide in favor of it.*

Response to Referee 2: We thank the Referee for the useful technical discussion. We accept that there are several important open challenges that need to be addressed on the road to scale-up of spin qubits. We agree too that there are many opportunities to explore solutions – this will be a very active area of research in the coming period. We believe the present work helps highlight both what is already possible and where new work can be focused to address the remaining challenges.

Referee 3

The authors have responded to all of my comments. Almost all of my comments are responded satisfactorily. Regarding the title mentioned in the response to "1. Temperature of the qubit chip may be ...", I prefer the authors' new suggestion "Spin Qubits with Scalable microwatt CMOS Control" to the original title. I still concern the authors' response to my comment "2. Influence of electrical noise from the cryo-CMOS ..." as follows:

1. Regarding "Lastly, we also remark ...", please elaborate the comparison in the manuscript with citing publications. If this comparison is appropriate, the smallness of noises contributed by the CMOS in the authors' claim is plausible.

2. While the authors mention "electron temperatures of order 200 mK" citing Ref. 7, I could not find descriptions about electron temperatures with powering up the CMOS device in Ref. 7. Unless this is my overlooking, this electron temperature value is unpublished and the authors should not use this to reinforce their argument without showing the supporting data.

Overall, I will be able to agree with the publication in Nature after the authors respond to my concern.

Response to Referee 3:

We thank the Referee for their careful reading our paper and insightful feedback provided. We are happy to adjust the title, as suggested by the Editor and Referee. We have now also modified the manuscript to include a comment about how the data compare to similar experiments without CMOS control, including citation to these publications.

C. H. Yang et al. Nature 580, 350 (2020),

W. Huang et al. Nature 569, 532, (2019).

In the 3rd paragraph in the Discussion section the sentence now reads:

“It is perhaps surprising then that we observe the CMOS chip to have only a small impact on qubit performance relative to previous experiments with room temperature control [17, 19].”

Lastly, on the question of electron temperatures measured previously, our comment was in the context of our response to the Referee – we do not refer to this unpublished data in the manuscript. There is the option to publish that dataset now here, although it would appear very out of context since it is on a GaAs quantum dot device (as described in ref. 7), rather than a silicon qubit and in a very different packaging arrangement (see Ref. 7). We feel this would likely be confusing to now bundle this old data set with the present paper. Instead, we intend to report such detailed thermal measurements in a future technical publication that will focus specifically on electron temperature management.